# Pericyte signaling via soluble guanylate cyclase shapes the vascular niche and microenvironment of tumors

Jing Zhu [1,2,9], Wu Yang [1,2,9 ✉], Jianyun Ma [2,3,9], Hao He [1,2,9], Zhen Liu [2,3,4,9], Xiaolan Zhu [1,2], Xueyang He [1,2], Jing He [1,2], Zhan Chen [5], Xiaoliang Jin [6], Xiaohong Wang [7], Kaiwen He [1,2,8], Wu Wei [3,4 ✉] & Junhao Hu [1,2,8 ✉]

## Abstract

**Pericytes and endothelial cells (ECs) constitute the fundamental components of blood vessels. While the role of ECs in tumor angiogenesis and the tumor microenvironment is well appreciated, pericyte function in tumors remains underexplored. In this study, we used pericyte-specific deletion of the nitric oxide (NO) receptor, soluble guanylate cyclase (sGC), to investigate via single-cell RNA sequencing how pericytes influence the vascular niche and the tumor microenvironment. Our findings demonstrate that pericyte sGC deletion disrupts EC–pericyte interactions, impairing Notch-mediated intercellular communication and triggering extensive transcriptomic reprogramming in both pericytes and ECs. These changes further extended their influence to neighboring cancer-associated fibroblasts (CAFs) and tumor-associated macrophages (TAMs) through paracrine signaling, collectively suppressing tumor growth. Inhibition of pericyte sGC has minimal impact on quiescent vessels but significantly increases the vulnerability of angiogenic tumor vessels to conventional anti-angiogenic therapy. In conclusion, our findings elucidate the role of pericytes in shaping the tumor vascular niche and tumor microenvironment and support pericyte sGC targeting as a promising strategy for improving anti-angiogenic therapy for cancer treatment.**

**Keywords** Soluble Guanylate Cyclase; Pericyte; Vascular Niche; Tumor Microenvironment
**Subject Categories** Cancer; Vascular Biology & Angiogenesis

## Introduction

Pericytes and the endothelial cells they ensheath are fundamental components that constitute blood vessels (Armulik et al, 2005). While their role in supporting tumor growth as conduits for oxygen and nutrient supply has long been recognized (Folkman, 1971; Weis and Cheresh, 2011), accumulating evidence now suggests that blood vessels actively release instructive signals that influence tumor cell dormancy, proliferation, and shape the immune landscape within the tumor microenvironment (Sobierajska et al, 2020). Previous studies have revealed that EC-derived LRG1 promotes tumor metastasis by preparing a perivascular metastatic niche in the lung (Singhal et al, 2021). EC-secreted CCL2 recruits CCR2+ macrophages to promote metastasis progression (Srivastava et al, 2014), while EC-derived Csf1 instructs macrophages to adopt an M2-like phenotype (He et al, 2012). In addition, fetal-like reprogrammed ECs interact with macrophages via the VEGF/NOTCH signaling to establish an immunosuppressive ecosystem in hepatocellular carcinoma (Sharma et al, 2020).

Pericytes are recruited by PDGFB signaling to support the growth of tumor-infiltrating blood vessels (Abramsson et al, 2003). However, only a few studies have explored their role in the tumor microenvironment (Picoli et al, 2021). A previous study has unveiled that low pericyte coverage in patients' tumors is associated with increased metastasis and poor prognosis (Yonenaga et al, 2005). This finding has been further substantiated in pre-clinical studies utilizing syngeneic 4T1 and spontaneous MMTV-PyMT breast cancer models, in which depletion of pericytes resulted in impaired blood vessel function, elevated intratumoral hypoxia, and enhanced metastasis (Cooke et al, 2012; Keskin et al, 2015). In addition to their crucial role as integral components of blood vessels, pericytes also provide environmental cues for tumor cell extravasation in the brain (Kienast et al, 2010). However, the majority of evidence comes from studies conducted on genetically modified mice models in which pericytes are either depleted or

[1]Interdisciplinary Research Center on Biology and Chemistry, Shanghai Institute of Organic Chemistry, Chinese Academy of Sciences, Shanghai, China. [2]University of Chinese Academy of Sciences, Beijing, China. [3]CAS Key Laboratory of Computational Biology, Shanghai Institute of Nutrition and Health, Chinese Academy of Sciences, Shanghai, China. [4]Lingang Laboratory, Shanghai, China. [5]Pathology Department, Cixi People's Hospital, Zhejiang, China. [6]Department of Ophthalmology, Ninth People's Hospital, Shanghai Jiao Tong University School of Medicine, Shanghai, China. [7]Department of Pharmacology and Tianjin Key Laboratory of Inflammation Biology, School of Basic Medical Sciences, Tianjin Medical University, Tianjin, China. [8]Shanghai Key Laboratory of Aging Studies, Shanghai, China. [9]These authors contributed equally as first authors: Jing Zhu, Wu Yang, Jianyun Ma, Hao He, Zhen Liu. ✉E-mail: wuyang@shsci.org; wuwei@lglab.ac.cn; jhhu@sioc.ac.cn

insufficiently recruited (Cooke et al, 2012; Keskin et al, 2015; Nisancioglu et al, 2010). Interestingly, a recent study reported that when β3 integrin is specifically depleted in pericytes, it does not impact pericyte recruitment or vessel density within the tumor. Instead, it results in increased expression of CCL2 in pericytes, which subsequently activates MEK1-ERK1/2-ROCK2 signaling in tumor cells, ultimately leading to accelerated tumor growth (Wong et al, 2020). Nevertheless, the complex interplay between pericytes and other cells within the tumor microenvironment remains largely unexplored.

NO-sGC signaling is recognized as the key signaling mediating the crosstalk between ECs and vascular smooth muscle cells for vascular tone regulation (Rees et al, 1989; Stasch et al, 2011). The expression of sGC in pericytes has only recently been reported (Bettaga et al, 2015; Friebe and Englert, 2022; Fukutani et al, 2009; He et al, 2023; He et al, 2016; Yang et al, 2021); nevertheless, its exact role in pericytes remains incompletely elucidated. Our recent findings revealed that NO-sGC represents one of the most significantly downregulated EC–pericyte crosstalk in the lung, primarily due to the marked decrease in sGC levels following an inflammatory challenge. This leads to pericyte detachment and increased vascular permeability. Moreover, pharmacological activation of sGC inhibits cytoskeleton reorganization and suppresses pericyte detachment, thereby preserving pericyte ensheathment and restoring vascular integrity (He et al, 2023). Notably, previous research demonstrated that EC-expressed eNOS and its product NO play an essential role in pericyte recruitment during tumor development (Kashiwagi et al, 2005). However, the broader impact of NO-sGC signaling on the tumor microenvironment requires further investigation.

In this study, we uncover the crucial role of pericytes in supporting tumor angiogenesis and shaping the tumor microenvironment. Employing single-cell RNA sequencing, we conducted comprehensive profiling of tumors with or without pericyte-specific sGC deletion. Our investigation reveals that the disruption of sGC signaling in pericytes results in compromised EC–pericytes interactions. This disruption not only triggers transcriptomic reprogramming within the vascular niche through Notch-dependent mechanisms, resulting in impaired tumor angiogenesis but also exerts a paracrine influence on neighboring CAFs and TAMs within the complex tumor microenvironment. Furthermore, our study highlights that pericyte sGC ablation enhances the responsiveness of tumor vessels to anti-angiogenic treatment, while sparing quiescent vessels from undesired side effects. These findings underscore the potential of pericyte targeting as a promising avenue for future clinical exploitation.

# Results

## sGC is expressed explicitly in tumor vascular pericytes

Previous studies have demonstrated that the NO receptor sGC is specifically expressed in the pericytes of healthy vessels in various organs, including the brain, retina, lung, liver, and muscle, with diverse functions ranging from maintaining vascular integrity to contributing to tissue fibrosis (Bettaga et al, 2015; Friebe and Englert, 2022; Fukutani et al, 2009; He et al, 2023; He et al, 2016; Yang et al, 2021). However, the expression pattern and function of

sGC in tumors have not been characterized. To address this, we inoculated Lewis lung carcinoma (LLC) into *sGC-EGFP* reporter mice (Fig. EV1A). Fluorescence microscopy analysis revealed that EGFP-expressing cells were positive for the pericyte-specific markers Desmin and NG2 and resided in proximity to CD31-positive EC (Fig. EV1B,C). In addition, 3D reconstruction of tumor vessels demonstrated the tight wrapping of capillaries by EGFP-expressing pericytes (Fig. EV1D). These data demonstrated that sGC is specifically expressed in tumor pericytes.

## Pericyte-specific sGC inactivation impairs tumor blood vessel stability and inhibits tumor growth

To investigate the impact of pericyte-specific sGC inactivation during tumor development, we bred *Gucy1b1flox/flox* (designated as *sGCCtr*) mice with *Cspg4-CreERT2* mice to generate *Cspg4-CreERT2::Gucy1b1flox/flox* (designated as *sGCΔPC*) mice, enabling the specific deletion of the catalytic β1 subunit of sGC in pericytes after administering five consecutive doses of Tamoxifen at the age of 4 weeks (Fig. 1A; Appendix Fig. S1A). The deletion efficiency of GUCY1B1 expression was validated using western blot analysis and immunofluorescent staining (Appendix Fig. S1B,C). Importantly, examination of brain vessels demonstrated that sGC inactivation in healthy adult mice had no noticeable impact on pericyte ensheathment of capillaries and vascular integrity even after 14 weeks of sGC inactivation (Appendix Fig. S1D,E), thus establishing a consistent baseline for evaluating tumor growth.

We then inoculated LLC tumors into *sGCCtr* and *sGCΔPC* littermates at the age of 8 weeks (Fig. 1A). Two weeks post inoculation, tumors were dissected for histological analysis. The growth of LLC tumors in *sGCΔPC* mice was significantly reduced compared to that in *sGCCtr* mice (Fig. 1B–D). To confirm that the reduced tumor growth in *sGCΔPC* mice was not confined to LLC tumors alone, we extended our investigation with additional syngeneic melanoma (B16F10) and breast cancer (EO771) models, which were inoculated subcutaneously and orthotopically, respectively, into both *sGCCtr* and *sGCΔPC* mice. Consistently, the growth of both B16F10 and EO771 tumors was markedly reduced in *sGCΔPC* mice (Appendix Fig. S2A–F). These compelling data indicated that NO-sGC signaling in pericytes plays an indispensable role in tumor growth.

To delineate the impact of pericyte-specific sGC inactivation on tumor vasculature, we sectioned LLC tumors and stained them with EC-specific marker CD31 and pericyte-specific markers NG2 and Desmin. Quantification of CD31+ blood vessels in LLC tumors revealed a significant decrease in vessel density in the *sGCΔPC* group (Fig. 1E). High-resolution confocal microscopy analysis demonstrated that the Desmin- and NG2-positive pericytes tightly enveloped the capillary endothelium in the *sGCCtr* LLC tumors. Surprisingly, we observed that a proportion of pericytes in the *sGCΔPC* LLC tumors were detached from the endothelium, significantly reducing pericyte coverage (Figs. 1F and EV2A). Consequently, the integrity of blood vessels in *sGCΔPC* LLC tumors was markedly impaired, as evidenced by the accumulation of extravasated albumin and Ter119-positive erythrocytes (Figs. 1G and EV2B). To evaluate vascular function subsequent to sGC deletion, we perfused the tumors with lectin and observed a profound reduction in the perfusion rate of blood vessels within *sGCΔPC* LLC tumors compared to those in *sGCCtr* LLC tumors

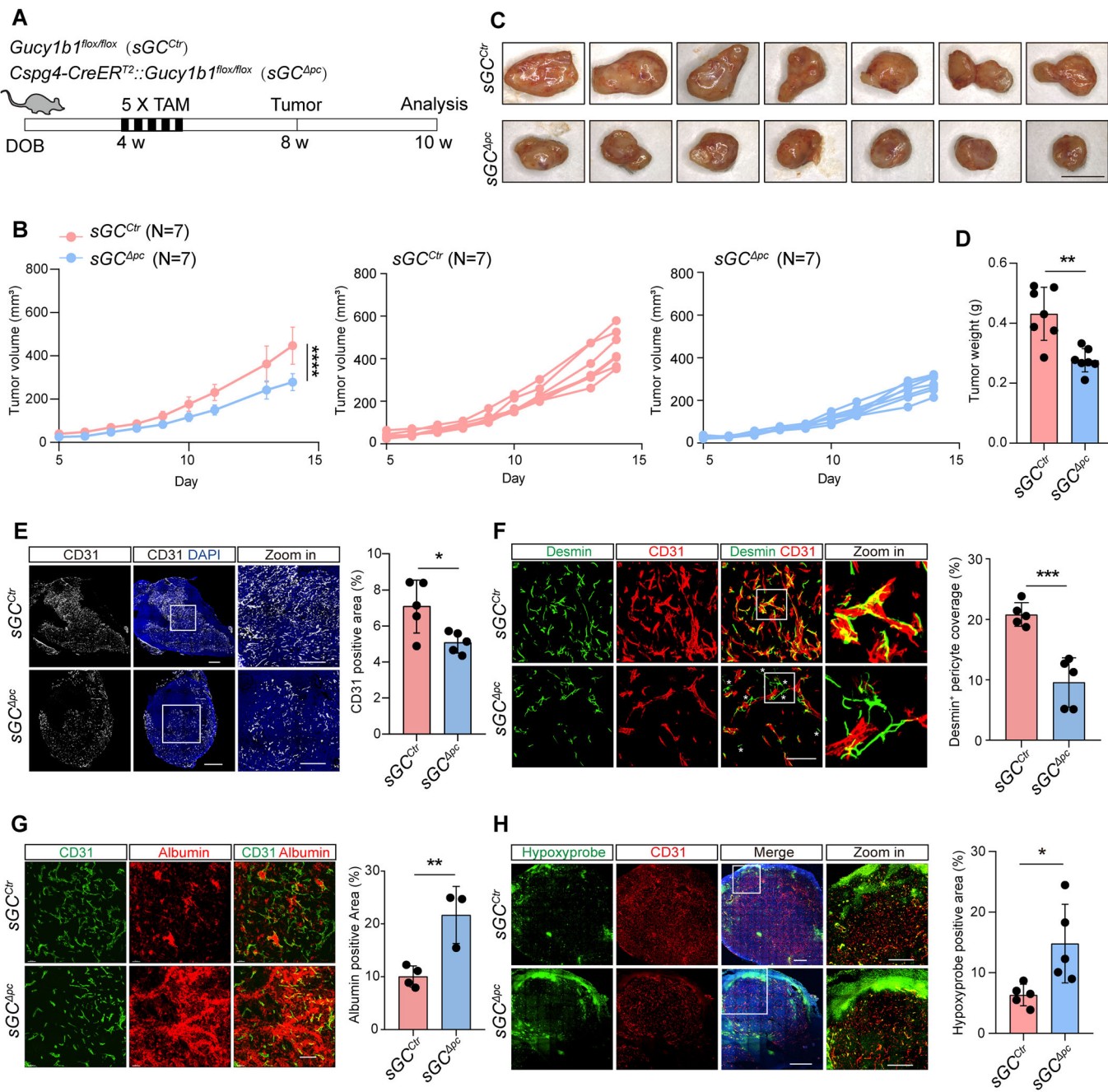

**Figure 1. Pericyte-specific sGC inactivation impairs tumor blood vessel and tumor growth.**

(A) Schematic depiction of the experimental design. Both *sGC^ctr^* and *sGC^ΔPC^* mice received five doses of tamoxifen at 4 weeks of age and were subcutaneously inoculated with LLC cells at 8 weeks of age. Tumors analysis were conducted 2 weeks post inoculation. (B) Growth curves of LLC tumors in *sGC^ctr^* and *sGC^ΔPC^* mice (left). Individual tumor growth curve for *sGC^ctr^* (middle) and *sGC^ΔPC^* (right) mice. Data presented as mean ± SD, with *n* = 7 mice per group. (C) Macroscope images of LLC tumors isolated from *sGC^ctr^* and *sGC^ΔPC^* mice. Scale bar, 10 mm. (D) Tumor weights of LLC tumors isolated from *sGC^ctr^* and *sGC^ΔPC^* mice. Data presented as mean ± SD, with *n* = 7 mice per group. (E) Representative fluorescence images of CD31 and DAPI-stained tumor sections from *sGC^ctr^* and *sGC^ΔPC^* mice. Plot depicting CD31-positive vessel area. Data presented as mean ± SD, with *n* = 5 mice per group. Scale bars, 1 mm (CD31 DAPI); 500 μm (Zoom-in). (F) Representative fluorescence images of CD31 and Desmin-stained tumor sections from *sGC^ctr^* and *sGC^ΔPC^* mice. Plot showing the percentage of vessels covered by Desmin-positive pericytes. Data presented as mean ± SD, with *n* = 5 mice per group. Scale bar, 100 μm. (G) Representative fluorescence images of CD31 and albumin-stained tumor sections from *sGC^ctr^* and *sGC^ΔPC^* mice. Plot showing the albumin-positive area within tumors. Data presented as mean ± SD, with *n* = 3–4 mice per group. Scale bar, 100 μm. (H) Representative fluorescence images of Hypoxyprobe, CD31, and DAPI-stained tumor sections from *sGC^ctr^* and *sGC^ΔPC^* mice. Plots indicating the hypoxyprobe-positive area within tumors. Data presented as mean ± SD, with *n* = 5 mice per group. Scale bars, 1 mm (Merge); 500 μm (Zoom-in). Statistical significance assessed using two-way ANOVA test with Tukey test (B) or two-tailed Student's *t* test (D–H). \*P < 0.05; \*\*P < 0.01; \*\*\*P < 0.001; \*\*\*\*P < 0.0001. Source data are available online for this figure.

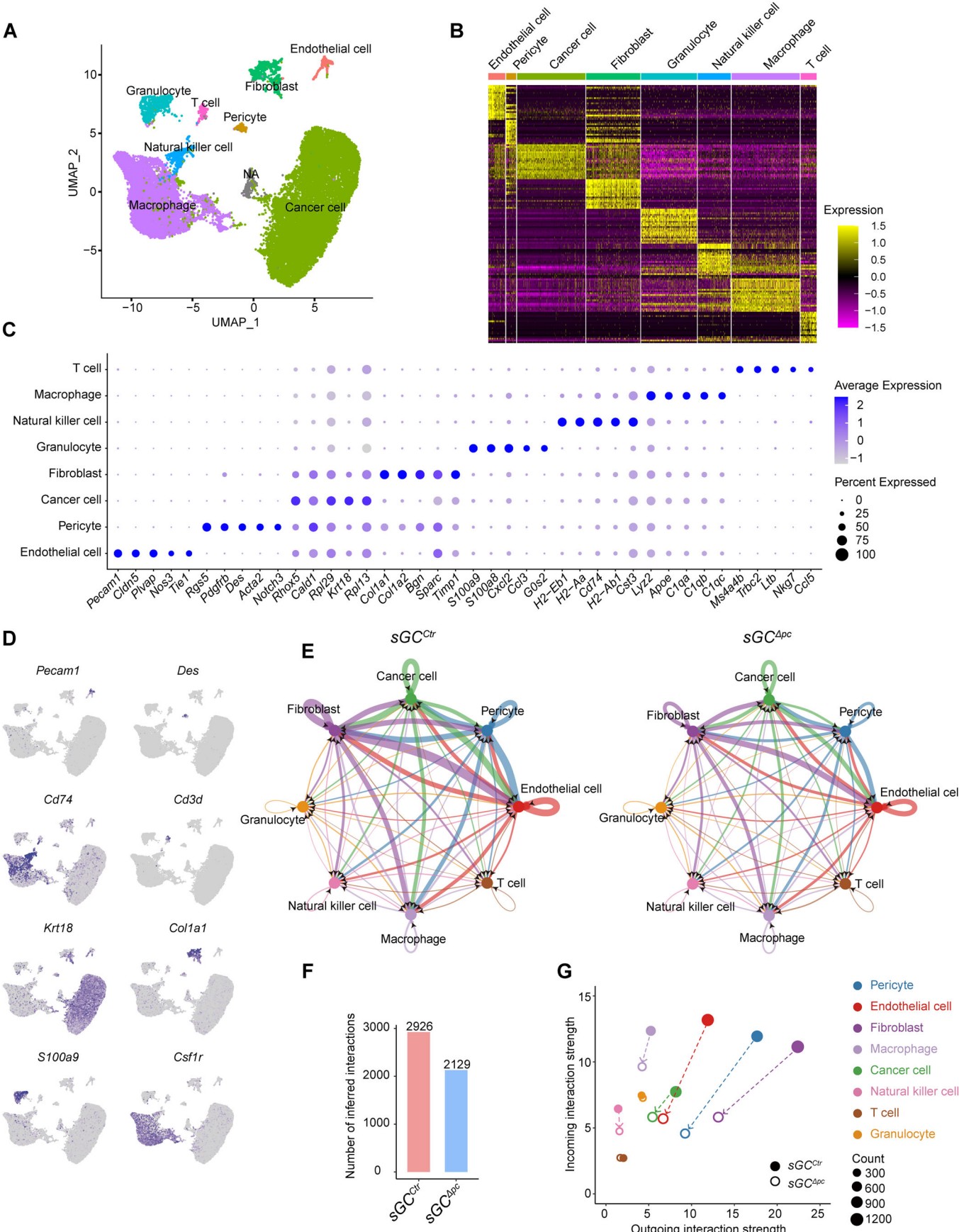

**Figure 2.   Altered intratumoral communications upon pericyte-specific sGC deletion.**

(**A**) UMAP visualization of 23383 single-cell transcriptomes combined from LLC tumors of *sGC^Ctr* (11,708) and *sGC^ΔPC* mice (11,675). (**B**) Heatmap displaying the top 20 marker genes for each cell cluster, arranged based on adjusted *P* values determined by the Wilcoxon rank-sum test. (**C**) Dot plots presenting normalized expression levels of corresponding markers of each cluster within tumors. (**D**) Distinct expression of cell marker genes superimposed on the UMAP plots from (**A**). (**E**) Circle plots depicting cell–cell interaction within tumors. Arrows indicate the direction of ligand-to-receptor interaction, while edge thickness denotes the cumulative weighted pathways between cell populations. (**F**) Numbers of total inferred cell–cell interactions in tumor. (**G**) Plot showing the shift of incoming and outgoing interaction strength for each cell population within tumors following pericyte-specific sGC deletion.

(Fig. EV2C). Furthermore, immunofluorescence analysis revealed a higher hypoxiaprobe signal in *sGC^ΔPC* LLC tumors compared to *sGC^Ctr* LLC tumors, indicating elevated intratumoral hypoxia (Fig. 1H). Similar findings were observed in B16F10 and EO771 tumors inoculated into *sGC^ΔPC* mice, showing decreased vessel density, reduced pericyte coverage, and impaired vascular integrity (Appendix Fig. S3A–D). These data indicate that NO-sGC signaling plays an important role in controlling tumor vessel function and tumor development.

## Single-cell transcriptomic analysis reveals altered intratumoral communications following pericyte-specific sGC deletion

To gain a comprehensive understanding of the direct impact of sGC deletion on tumor pericytes and its indirect effects on EC, cancer cells, and other stromal cells within the tumor microenvironment, we conducted single-cell transcriptomic analysis on *sGC^Ctr* and *sGC^ΔPC* LLC tumors. After stringent quality control, we integrated transcriptomes of 23,383 high-quality single cells (11,078 cells from *sGC^Ctr* LLC tumors and 11,675 cells from *sGC^ΔPC* LLC tumors) into a unified dataset. Using unsupervised clustering and Uniform Manifold Approximation and Projection (UMAP) visualization, we classified these cells into eight major cell clusters. Cluster-specific differential gene expression analysis, guided by established cell-type specific markers, allowed us to define these clusters as ECs, pericytes, fibroblasts, macrophages, T cells, granulocytes, natural killer cells, and cancer cells (Fig. 2A–D). Among these clusters, LLC cells (59.055%) constituted the largest cell population, followed by macrophages (29.104%) as the most abundant stromal population. In contrast, ECs and pericytes accounted for only a small percentage of the analyzed cells (1.081% and 0.647%, respectively) (Appendix Fig. S4A). To rule out the possibility that the low presentation of EC and pericytes may be attributed to the cell dissociation and cell capturing processes during single-cell analysis, we inoculated LLC tumors into *sGC-EGFP* mice and examined the proportions of ECs and pericytes using fluorescence-activated cell sorting (FACS) analysis. Consistent with the findings from the single-cell analysis, FACS analysis identified 1.06% ECs (CD41⁻CD45⁻Ter119⁻CD31⁺) and 0.68% pericytes (CD41⁻CD45⁻Ter119⁻EGFP⁺) in LLC tumors (Appendix Fig. S4B).

Considering the pivotal role of intercellular crosstalk between cancer cells and stromal cells in shaping the tumor microenvironment and driving tumor growth (Hinshaw and Shevde, 2019), we employed CellChat (Jin et al, 2021)—a tool that infers intercellular communication based on the expression of ligands, receptors and their cofactors in different cell populations—to examined the global alterations in cell–cell communications within LLC tumors

following pericyte-specific sGC deletion (Fig. 2E). Analysis unveiled a substantial reduction in the total number of inferred intercellular interactions, decreasing from 2926 intercellular interactions in *sGC^Ctr* tumors to 2129 intercellular interactions in *sGC^ΔPC* tumors (Fig. 2F). Particularly, despite pericytes, ECs, and CAFs constituting a minor fraction of cells within LLC tumors, they exhibited the most significant reductions in both incoming and outgoing interaction strength subsequent to sGC deletion in pericytes (Fig. 2G). In addition, the incoming and outgoing interactions in cancer cells, macrophages, and natural killer cells were also affected by pericyte-specific sGC deletion. In contrast, the crosstalk between T cells and granulocytes with other cells was minimally affected following pericyte-specific sGC deletion. Collectively, the remarkable changes in the intratumoral crosstalk suggest that pericytes are important players in controlling the tumor microenvironment.

## Loss of pericyte sGC reprograms transcriptome in both ECs and pericytes and impairs EC–pericyte crosstalk within the vascular niche

To gain insight into the altered function of ECs and pericytes within the vascular niche of LLC tumors following pericyte-specific sGC deletion, we analyzed transcriptomic changes using the Wilcoxon rank-sum test within the Seurat package, performing differential expression analysis with a uniform cutoff in both cell types. We observed a remarkable gene expression shift, with 844 genes in pericytes and 862 genes in ECs downregulated upon pericyte-specific sGC deletion. In contrast, only 144 genes in pericytes and 147 genes in ECs were upregulated (Fig. 3A). Gene ontology (GO) biological pathway analysis displayed that the upregulated genes in pericytes were enriched for immune cell chemotaxis and migration, indicating a pro-inflammatory state following sGC deletion (Fig. 3B). These results align with previous studies showing that sGC is required to suppress the expression of inflammatory cytokines in the pericytes of the lung and liver (Flores-Costa et al, 2018; He et al, 2023; Yang et al, 2021). The downregulated genes in pericytes were involved in an extracellular matrix organization and cell–matrix adhesion (Fig. 3B), which contribute to the pericyte detachment from the endothelium. Apart from *Gucy1b1* and *Notch3*, the expression of classical pericyte markers, including *Desmin, Rgs5*, and *Pdgfrb*, was only mildly affected (Appendix Fig. S5A). Notably, in ECs of *sGC^ΔPC* LLC tumors, the downregulated genes were notably enriched in functions pivotal to endothelial cell migration, angiogenesis, and the VEGF signaling pathway (Fig. 3B). Detailed analysis revealed that while the expression of common EC markers such as CD31, Claudin-5, and collagen-4 remained largely unaffected (Appendix Fig. S5B–D), the mRNA and protein levels of *Pdgfb, Tek, Nos3*, and *Kdr*, underwent substantial downregulation (Fig. 3C–F). These

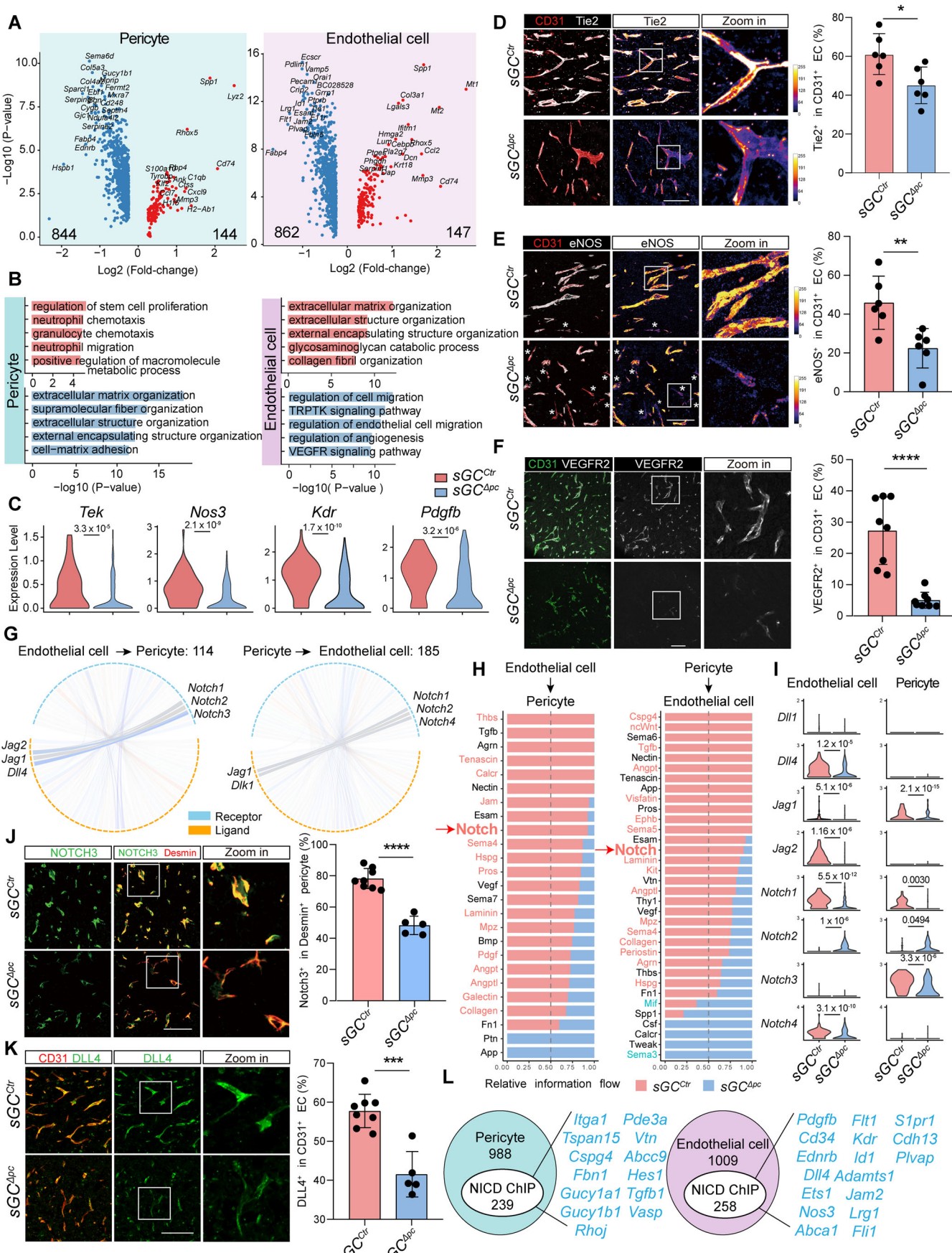

◄

**Figure 3.  Pericyte sGC deletion alters gene transcription and modifies EC–pericyte interaction within the vascular niche.**

(A) Volcano plots showing the DEGs in pericytes and ECs between *sGC^ctr* and *sGC^ΔPC* mice. DEGs were identified utilizing the Wilcoxon rank-sum test. (B) GO analysis of DEGs in pericytes and ECs. (C) Violin plots displaying expression levels of *Tek, Nos3, Kdr,* and *Pdgfb* in ECs. (D) Representative fluorescence images showing CD31 and Tie2 stained tumor sections from *sGC^ctr* and *sGC^ΔPC* mice. Plots showing the proportion of Tie2^+ in CD31^+ ECs. Data presented as mean ± SD, with *n* = 6 mice per group. Scale bar, 100 μm. (E) Representative fluorescence images showing CD31 and eNOS stained tumor sections from *sGC^ctr* and *sGC^ΔPC* mice. Plots showing the proportion of eNOS^+ in CD31^+ ECs. Asterisks denote eNOS negative ECs. Data presented as mean ± SD, with *n* = 6 mice per group. Scale bar, 100 μm. (F) Representative fluorescence images showing CD31 and VEGFR2 stained tumor sections from *sGC^ctr* and *sGC^ΔPC* mice. Plots showing the proportion of VEGFR2^+ in CD31^+ ECs. Data presented as mean ± SD, with *n* = 8 mice per group. Scale bar, 100 μm. (G) Circle plots visualizing 144 EC–pericyte interactions and 185 pericyte-EC interactions identified in tumors. (H) Bar chart illustrating the proportions of signaling pathways involved in EC–pericyte crosstalk in in *sGC^ctr* and *sGC^ΔPC* mice. Pathways with significantly higher information flow in *sGC^ctr* mice are highlighted in red font, those with significantly higher information flow in *sGC^ΔPC* mice are highlighted in blue font, and black fond denotes no significant differences. (I) Violin plots showing expression levels of *Dll1, Dll4, Jag1, Jag2, Notch1, Notch2, Notch3,* and *Notch4* in *sGC^ctr* and *sGC^ΔPC* mice tumors. (J) Representative fluorescence images of NOTCH3 and Desmin-stained tumor sections from *sGC^ctr* and *sGC^ΔPC* mice. Plots showing the proportion of NOTCH3^+ in Desmin^+ pericytes. Data presented as mean ± SD, with *n* = 5–8 mice per group. Scale bar, 100 μm. (K) Representative fluorescence images of CD31 and DLL4 stained tumor sections from *sGC^ctr* and *sGC^ΔPC* mice. Plots showing the proportion of DLL4^+ in CD31^+ ECs. Data presented as mean ± SD, with *n* = 5–8 mice per group. Scale bar, 100 μm. (L) Enrichment analysis of NICD ChIP-seq dataset (GSE34954) displaying NICD presence at the promoter regions of 239 DEGs of pericytes and 258 DEGs of endothelial cells. Statistical significance assessed using unpaired two-sample Wilcoxon test (C, I), two-tailed Student's *t* test (D–F, J, K). *P < 0.05; **P < 0.01; ***P < 0.001; ****P < 0.0001. Source data are available online for this figure.

results explicate the underlying mechanisms for the diminished pericyte coverage and reduced vessel density in *sGC^ΔPC* LLC tumors.

Given the close proximity and reciprocal influence of pericytes and ECs (Armulik et al, 2005), we investigated the changes in crosstalk between pericytes and ECs. Comparative interactome analysis revealed significant alterations in 114 EC-to-pericyte interactions and 185 pericyte-to-EC interactions in LLC tumors following pericyte-specific sGC inactivation (Fig. 3G). Notably, the Notch signaling pathway, known for its essential role in controlling blood vessel development and maturation (Roca and Adams, 2007), exhibited significant impairment in both EC-to-pericyte and pericyte-to-EC crosstalks in *sGC^ΔPC* LLC tumors (Fig. 3G,H). Moreover, a comprehensive examination of Notch signaling across all cell types within LLC tumors revealed that, while Notch signaling mediates intercellular crosstalk between multiple cell types, the most robust Notch signaling activity occurred between ECs and pericytes (Fig. EV3A), further emphasizing the pivotal role of Notch signaling in regulating blood vessel function. The disruption of Notch signaling between EC and pericytes can be partially attributed to the observed pericyte detachment in *sGC^ΔPC* LLC tumors (Fig. 1G), as the activation of Notch signaling relies on direct cell–cell contact. However, the impaired Notch signaling-mediated crosstalk may also arise from altered expression of Notch ligands and receptors. Indeed, ECs of *sGC^ΔPC* LLC tumors showed significantly reduced mRNA levels of Notch ligands *Dll4* and *Jag2*, as well as Notch receptors *Notch1* and *Notch4*, compared to their expression in ECs of *sGC^Ctr* LLC tumors. Similarly, pericyte in *sGC^ΔPC* LLC tumors exhibited reduced expression levels of *Jag1, Notch1,* and *Notch3* (Fig. 3I). Immunofluorescence staining validated the decreased DLL4 expression in ECs and NOTCH3 expression in pericytes in *sGC^ΔPC* LLC tumors (Fig. 3J,K), confirming the disruption of Notch crosstalk between ECs and pericytes upon pericyte-specific sGC inactivation.

Since Notch regulates various target gene expressions via the releases of its intracellular domain (NICD), we, therefore, assessed the involvement of Notch signaling in the observed transcriptomic alterations within both ECs and pericytes. To achieve this, we performed a comparative analysis by correlating the altered gene sets of ECs and pericytes with previously published NICD ChIP-seq data. This analysis revealed a substantial proportion of the altered genes in both cell types were potentially regulated by NICD, as

evidenced by the highly enriched binding sites within the genomic sequence of those genes. Notably, we noticed that a subset of Notch-regulated genes was either crucial for cell identities or angiogenic function, such as *Gucy1a1, Gucy1b1, and Cspg4* in pericytes and *Pdgfb, Kdr, Nos3, Cd34, Endrb*, and *Plvap* in ECs (Figs. 3L and EV3B). To validate this finding, we overexpressed NICD in both ECs and pericytes and analyzed the expression of the predicted Notch-regulated genes. The result demonstrated that NICD overexpression led to elevated expression levels of *PLVAP, FLT1, ENDRB*, and *ADAMTS1* in ECs, and significantly increased expression levels of *GUCY1A1, GUCY1B1, ABCC9, VASP, TSPAN15*, and *TGFB1* in pericytes. expression (Fig. EV3C–E). These findings provide compelling evidence that Notch signaling plays a pivotal role in orchestrating gene expression in both ECs and pericytes. The targeted deletion of sGC in pericytes disrupts their attachment to ECs, subsequently impeding Notch signaling and, as a consequence, triggering extensive transcriptomic reprogramming in both cell types within the vascular niche.

## Pericyte sGC deletion promotes the expansion of tumor-inhibiting CD105^neg CAFs

Single-cell analysis revealed the most prominently expanded cell population following pericyte-specific sGC deletion within the tumors was CAFs (Fig. 4A,B). Additional experiments using FACS analysis and immunostaining with antibodies against fibroblast-specific markers Podoplanin and PDGFRα confirmed the remarkable increase of CAFs in LLC tumors (Appendix Fig. S6A,B). CAFs in tumor stroma play pivotal roles in tumor development (Biffi and Tuveson, 2021; Caligiuri and Tuveson, 2023). Particularly, recent studies utilizing single-cell analysis have revealed the remarkable heterogeneity among CAFs, further highlighting the complexity and function diversity of CAFs within the context of cancer (Lavie et al, 2022). Through differential gene expression analysis, CAFs in LLC tumors were classified into three distinct clusters (Fig. 4C,D), with well-established fibroblast-specific markers *Col1a1, Col1a2, Timp1*, and *Dcn* expressed in all three clusters (Fig. 4E). Cluster-1 CAFs specifically expressed proliferation-associated genes (*Mki67, Cdk1*, and *Top2a*) (Fig. 4F). However, the cluster-2 CAFs in LLC tumors exhibited a mixed gene signature, showing features of both inflammatory CAF (iCAF) and myofibroblastic CAF (myCAF)

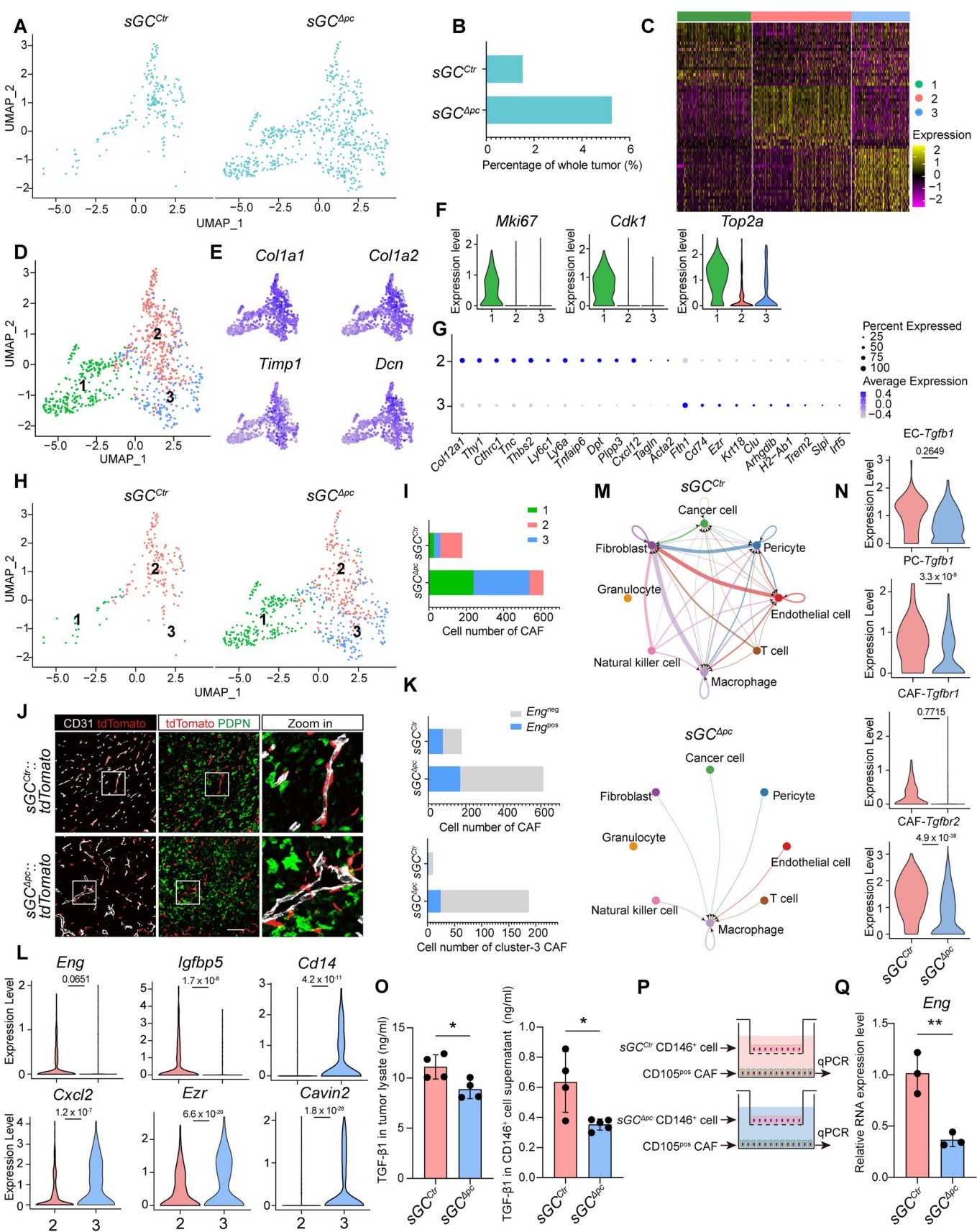

**Figure 4. Pericyte sGC deletion promotes tumor-inhibiting CD105ⁿᵉᵍ CAFs expansion.**

(A) UMAP visualization of CAFs from combined fibroblasts of sGCᶜᵗʳ and sGCᐩᴬᴾᶜ tumors. (B) Proportion of CAFs within the whole tumor. (C) Heatmap displaying the top 20 marker genes for each cell cluster, arranged based on adjusted *P* values determined by the Wilcoxon rank-sum test. (D) UMAP plots highlight distinct fibroblast clusters. (E) Fibroblast pan marker genes overlaid on UMAP plots of (D). (F) Violin plots showing expression levels of *Mki67*, *Cdk1*, and *Top2a* in cluster 1. (G) Dot plots presenting normalized expression levels of corresponding markers of cluster 2 and cluster 3. (H) UMAP visualization of three clusters after pericyte sGC deletion. (I) Cell numbers of three CAF clusters within CAFs. (J) Representative fluorescence images showing CD31, tdTomato, and Podoplanin-stained LLC tumor sections from sGCᶜᵗʳ::tdTomato and sGCᐩᴬᴾᶜ::tdTomato mice. Scale bar, 100 μm. (K) Cell numbers of Engᵖᵒˢ and Engⁿᵉᵍ cells in fibroblasts. (L) Violin plots showing expression levels of CD105ᵖᵒˢ and CD105ⁿᵉᵍ CAFs marker genes in cluster-2 and cluster-3 CAFs. (M) Circle plot of the TGFβ signaling. Arrows indicate ligand-to-receptor direction, while edge thickness denotes the cumulative weighted pathways between cell populations. (N) Violin plots showing expression levels of *Tgfb1*, *Tgfbr1*, and *Tgfbr2* in ECs, pericytes, and fibroblasts. (O) Plots revealing active TGF-β1 levels in whole-tumor lysates and CD146⁺ cell supernatants of sGCᶜᵗʳ and sGCᐩᴬᴾᶜ tumors, measured using ELISA. Data presented as mean ± SD, with n = 4–5 mice per group. (P) Schematic depiction of the experimental design. CD105ᵖᵒˢ CAFs were placed in the transwell bottom chamber, CD146⁺ cells in the upper chamber. (Q) Plot illustrating *Eng* expression levels of CD105ᵖᵒˢ CAFs by qPCR after 48 h co-culture. Data presented as mean ± SD, with n = 3 mice per group. Statistical significance assessed using unpaired two-samples Wilcoxon test (L, N), two-tailed Student's *t* test (O, Q). *P < 0.05; **P < 0.01. Source data are available online for this figure.

(Affo et al, 2021), as evidenced by the expression of inflammatory response-associated genes (*Ly6a*, *Cxcl12*, etc.) and cell migration and ECM-associated genes (*Col12a1*, *Tnc*, *Tagln*, *Acta2*, etc.). Cluster-3 CAFs showed a gene signature of antigen-presenting CAF (apCAF) (Elyada et al, 2019), characterized by higher expression levels of *Cd74*, *H2-Ab1*, *Trem2*, etc. (Fig. 4G). Notably, the expansion of CAFs in sGCᐩᴬᴾᶜ LLC tumors was primarily contributed by the substantial increase in cluster-1 and cluster-3 CAFs, whereas the number of cluster-2 CAFs was mildly decreased (Fig. 4H,I).

Given the remarkable plasticity of pericytes to differentiate into various mesenchymal cell types, including CAFs, in both physiological and pathological contexts (Goritz et al, 2011; Hosaka et al, 2016; Volz et al, 2015), we sought to investigate whether the deletion of sGC would induce an aberrant transition of pericytes into fibroblasts. To address this question, we crossed sGCᶜᵗʳ and sGCᐩᴬᴾᶜ mice with *Rosa26-LSL-tdTomato* mice (designated as sGCᶜᵗʳ::tdTomato and sGCᐩᴬᴾᶜ::tdTomato mice, respectively), allowing us to genetically label pericytes with tdTomato and trace their progeny within LLC tumors (Appendix Fig. S7A). Immunofluorescence imaging analysis revealed that all tdTomato-labeled cells within tumors from both sGCᶜᵗʳ::dtTomato and sGCᐩᴬᴾᶜ::tdTomato mice consistently expressed the pericyte marker Desmin (Appendix Fig. S7B). Interestingly, further staining of these tumor sections with CAF-specific marker Podoplanin demonstrated that none of the tdTomato-labeled pericytes were positive for Podoplanin, including those pericytes detached from blood vessels (Fig. 4J). These findings unequivocally demonstrated that deleting sGC in pericyte does not induce a pericyte-to-fibroblast transition within tumors.

Next, we analyzed the impact of pericyte-specific sGC deletion on CAFs by analyzing their gene expression signature. Interestingly, in addition to their apCAFs signature, we found the majority of cluster-3 CAFs in sGCᐩᴬᴾᶜ LLC tumors were absent of CD105 expression (Figs. 4K and EV4A). CD105 is encoded by the gene Endoglin (*Eng*) and is known to modulate TGFβ signaling by interacting with type-I and type-II TGFβ receptors (Cheifetz et al, 1992). To validate this finding, we performed FACS analysis and confirmed a significant increase in the proportion of CD105ⁿᵉᵍ CAFs in sGCᐩᴬᴾᶜ LLC tumors compared to those in sGCᶜᵗʳ LLC tumors (Fig. EV4B,C). While CD105ⁿᵉᵍ CAFs were presented in both cluster-2 and cluster-3 CAFs, the majority of CD105ⁿᵉᵍ CAFs fell within the cluster-3 category (Fig. 4K). This finding was further

substantiated through qPCR-based target gene analysis of sorted CAFs, in which CD105ᵖᵒˢ CAFs demonstrated elevated *Eng* expression along with cluster-2 markers *Igfbp5* and *CXCL14*, while CD105ⁿᵉᵍ CAFs displayed heightened levels of *Cd74*, *Cd14*, and *Cxcl2* expression, notably devoid of *Eng* expression (Figs. 4L and EV4D). Previous reports demonstrated that CD105ⁿᵉᵍ CAFs are highly tumor-suppressive in mouse and human tumors (Hutton et al, 2021). It has been demonstrated that the switch between CD105ⁿᵉᵍ CAFs and CD105ᵖᵒˢ CAFs is regulated by TGFβ signaling in the tumor microenvironment (Hutton et al, 2021; Watt and Morton, 2021). Indeed, treatment with TGFβ1 upregulated CD105 expression in CD105ⁿᵉᵍ CAFs isolated from LLC tumors (Fig. EV4E). These findings prompted us to compare the TGFβ signaling between sGCᶜᵗʳ LLC tumors and sGCᐩᴬᴾᶜ LLC tumors. Based on CellChat analysis, we observed a significant decrease in TGFβ-mediated intercellular communication in sGCᐩᴬᴾᶜ LLC tumors, particularly the transmission of TGFβ signaling from ECs and pericytes within the vascular niche to fibroblasts was almost completely abolished (Fig. 4M). The impairment of TGFβ activity in sGCᐩᴬᴾᶜ LLC tumors was further supported by a significant reduction in pSMAD3 levels, a key downstream effector of the TGFβ signaling pathway (Fig. EV4F). This disruption could be attributed to the downregulation of *Tgfb1* in both pericytes and ECs, as well as the reduced expression of its cognate receptors, *Tgfbr1* and *Tgfbr2*, in CAFs (Fig. 4N). To validate the ability of intratumoral ECs and pericytes to regulate CD105 expression in CAFs via TGFβ signaling, we co-culture CAFs with ECs and pericytes isolated from either sGCᶜᵗʳ LLC tumors or sGCᐩᴬᴾᶜ LLC tumors. Given that both ECs and pericytes within the vascular niche of LLC tumors expressed CD146 (Appendix Fig. S8A–C), which is in agreement with previous studies (Meng et al, 2021), we proceeded to isolate both cell types simultaneously using CD146 magnetic beads for following co-culture experiments. We first quantified secreted TGFβ1 levels using an ELISA assay and confirmed the significant reduction of TGFβ1 in the supernatants of ECs and pericytes isolated from sGCᐩᴬᴾᶜ LLC tumors as well as in the whole-tumor lysates of sGCᐩᴬᴾᶜ mice (Fig. 4O). Furthermore, when co-cultured with ECs and pericytes isolated from sGCᐩᴬᴾᶜ LLC tumors, CD105ᵖᵒˢ CAFs exhibited decreased CD105 expression compared to those co-cultured with EC and pericytes isolated from sGCᶜᵗʳ LLC tumors (Fig. 4P,Q), indicating a phenotypic switching to CD105ⁿᵉᵍ apCAFs. Collectively, the data demonstrated that sGC deletion decreased TGFβ1 expression in the vascular niche,

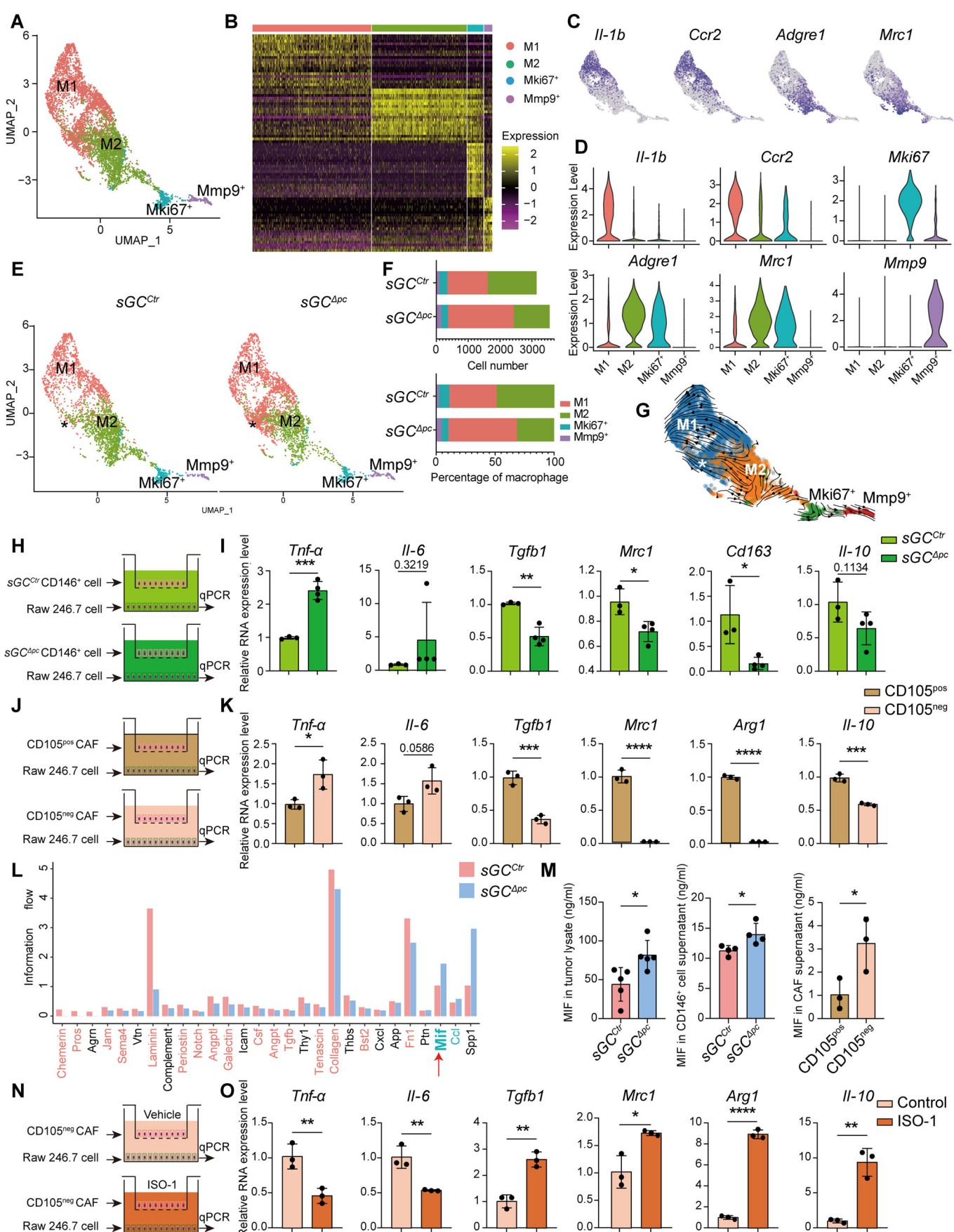

**Figure 5. Pericyte sGC deletion influences TAM polarization.**

(A) UMAP visualization of TAMs within $sGC^{ctr}$ and $sGC^{\Delta PC}$ tumors. (B) Heatmap displaying the top 20 marker genes for each cell cluster, arranged based on adjusted *P* values determined by the Wilcoxon rank-sum test. (C) Distinct expression of M1 and M2 marker genes overlaid on the UMAP plots of (A). (D) Violin plots showing expression levels of marker genes across different TAM sub-clusters. (E) UMAP visualization of four clusters after pericyte sGC deletion. Asterisk denotes the notable increase of M1-like TAMs. (F) Plots displaying cell numbers and proportions of the four TAM sub-clusters. (G) RNA velocity analysis showing transition potential among TAM sub-clusters. The asterisk denotes the newly emerged M1-like TAMs arise from the conversion of M2-like TAMs. (H) Schematic depiction of the experimental design. CD146⁺ cells were placed in the transwell upper chamber, Raw246.7 cells in the bottom chamber. (I) Plot showing gene expression levels of *Tnf-α*, *Il-6*, *Tgfb1*, *Mrc1*, *Cd163*, and *Il-10* in Raw246.7 cells after 48 h co-culture. Data presented as mean ± SD, with $n = 3$–4 mice per group. (J) Schematic depiction of the experimental design. CAFs were placed in the transwell upper chamber, Raw246.7 cells in the bottom chamber. (K) Plot showing gene expression levels of *Tnf-α*, *Il-6*, *Tgfb1*, *Mrc1*, *Arg1*, and *Il-10* in Raw246.7 cells after 48 h co-culture. Data presented as mean ± SD, with $n = 3$ replicates. (L) Comparison of the overall information flow of each signaling pathway between $sGC^{ctr}$ and $sGC^{\Delta PC}$ tumors. with ECs, pericytes and CAFs are signal senders, while TAMs are signal receiver. The pathways with greater information flow in $sGC^{ctr}$ or $sGC^{\Delta PC}$ were marked in cyan or red, respectively, while black indicates no significant differences. (M) Plots showing MIF levels in whole-tumor lysates, supernatants of CAFs, and CD146⁺ cell of $sGC^{ctr}$ and $sGC^{\Delta PC}$ mice, MIF concentration was determined using ELISA. Data presented as mean ± SD, with $n = 3$–5 mice per group. (N) Schematic depiction of the experimental design. CD105ⁿᵉᵍ CAFs were seeded in the transwell upper chamber, Raw246.7 cells in the bottom chamber. The culture medium was supplemented with vehicle or 100 µg/ml ISO-1. (O) Plot showing gene expression levels of *Tnf-α*, *Il-6*, *Tgfb1*, *Mrc1*, *Arg1*, and *Il-10* in Raw246.7 cells following 48 h co-culture. Data presented as mean ± SD, with $n = 3$ replicates. Statistical significance assessed using two-tailed Student's *t* test (I, K, M, O). *$P < 0.05$; **$P < 0.01$; ***$P < 0.001$; ****$P < 0.0001$. Source data are available online for this figure.

consequently reprogrammed adjacent CAFs into CD105ⁿᵉᵍ apCAFs, potentially contributing to the observed phenotype of reduced tumor growth.

## Pericyte sGC deletion interferes with TAM polarization

We next investigate whether TAMs, which constituted the largest stromal population (Fig. 2A; Appendix Fig. S4A), were affected by pericyte-specific sGC deletion. Single-cell analysis classified TAMs into four distinct populations based on their gene expression signatures: $Il1b^{hi}Ccr2^{hi}$ M1-like TAMs, $Adgre1^{hi}Mrc1^{hi}$ M2-like TAMs, $Mki67^+$ TAMs, and $Mmp9^+$ TAMs (Fig. 5A–D). It is well-established that M1-like TAMs exert tumor-suppressive effects, while M2-like TAMs promote tumor growth (Mills, 2012; Noy and Pollard, 2014). Interestingly, we observed an alteration in the ratio of M1-like to M2-like TAMs in $sGC^{\Delta PC}$ LLC tumors compared to $sGC^{ctr}$ LLC tumors, with a notable increase in the abundance of M1-like TAMs (Fig. 5E,F). Emerging evidence demonstrates that TAMs are extremely plastic and can be converted to each other according to environmental cues (Sica et al, 2008). To explore whether the increased M1-like TAM in $sGC^{\Delta PC}$ LLC tumors results from altered macrophage polarization, we performed single-cell RNA velocity analysis using scVelo dynamical model (Bergen et al, 2020). Analysis revealed that although most M1-like and M2-like TAMs shared a common origin, they exhibited divergent differentiation trajectories. Interestingly, the RNA velocity analysis suggested that the newly emerged M1-like TAMs in $sGC^{\Delta PC}$ LLC tumors potentially arise from the conversion of M2-like TAMs rather than direct differentiation from their shared origin (Fig. 5G).

To uncover the environmental cues influencing the shift of M2-like TAMs towards M1-like TAMs, we first assessed the interactions between TAMs and other cellular components within the tumor microenvironment. Our analysis unveiled a significant reduction in the crosstalk between TAMs and ECs, pericytes, and CAFs (Fig. EV5A). To ascertain whether ECs, pericytes, and CAFs directly influence macrophage polarization, we isolated those cell populations from $sGC^{ctr}$ or $sGC^{\Delta PC}$ LLC tumors and co-cultured with RAW264.7 mouse macrophages. Subsequently, we employed qPCR analysis to examine M1-like or M2-like gene signatures in RAW264.7 macrophages. We observed that ECs and pericytes from $sGC^{\Delta PC}$ LLC tumors potently upregulated the expression of M1-like

signature genes (*Tnfa* and *Il-6*) while strongly inhibiting the expression of M2-like signature genes (*Tgfb1*, *Mrc1*, *Cd163*, *Il-10*) in RAW264.7 macrophages when compared to ECs and pericytes isolated from $sGC^{ctr}$ LLC tumors (Fig. 5H,I). In addition, we found that CD105ⁿᵉᵍ CAFs, which were dramatically expanded in $sGC^{\Delta PC}$ LLC tumors, exerted a similar effect to that of ECs and pericytes from $sGC^{\Delta PC}$ LLC tumors, were more potent than CD105ᵖᵒˢ CAFs in promoting macrophage polarization towards an M1-like phenotype (Fig. 5J,K).

To identify the key signaling pathways governing macrophage polarization, we conducted a comparative analysis of pathways that impact TAMs in $sGC^{ctr}$ and $sGC^{\Delta PC}$ LLC tumors. Analysis suggested that MIF, an important regulator of macrophage polarization, was one of the significantly upregulated pathways in $sGC^{\Delta PC}$ LLC tumors (Fig. 5L). Moreover, examination of single-cell transcriptomic data revealed elevated expression levels of *Mif* in ECs, pericytes, and CAFs within $sGC^{\Delta PC}$ LLC tumors in comparison to their counterparts in $sGC^{ctr}$ LLC tumors (Fig. EV5B,C). These findings were further substantiated by ELISA analysis, which confirmed higher protein levels of MIF in $sGC^{\Delta PC}$ LLC tumors compared to $sGC^{ctr}$ LLC tumors (Fig. 5M). Further investigation of the supernatant from ECs and pericytes revealed that cells in the vascular niche of $sGC^{\Delta PC}$ LLC tumors secrete more MIF than those from $sGC^{ctr}$ LLC tumors. Similarly, CD105ⁿᵉᵍ CAFs secreted more MIF than CD105ᵖᵒˢ CAFs (Fig. 5M). Consistent with previous studies (Castro et al, 2017), our data demonstrated that direct exposure of macrophages to MIF induced an M2-to-M1 transition, as evidenced by the upregulation of M1-like gene signature (*Nos2*, *Il-6*, *Il-1b*, *Tnf-a*) and a concurrent reduction in M2-like gene signature (*Il-10*, *Tgfb1*, *Arg1*, *Cd163*, *Mrc1*) (Fig. EV5D). To further consolidate that the vascular niche and CAFs-derived MIF drives TAMs towards an M1-like polarization via MIF in $sGC^{\Delta PC}$ tumor, we treated RAW264.7 with ISO-1, a highly specific inhibitor of MIF tautomerase activity, alone or when co-cultured with ECs and pericytes or with CD105ⁿᵉᵍ CAFs. Remarkably, ISO-1 treatment alone did not exert any discernible influence on the expression of genes associated with either the M1 or M2-like signatures (Fig. EV5E,F). However, ISO-1 treatment completely abrogated the upregulation of genes linked to the M1-like signature and reversed the suppression of genes associated with the M2-like signature in RAW264.7 cells co-cultured with EC and pericytes

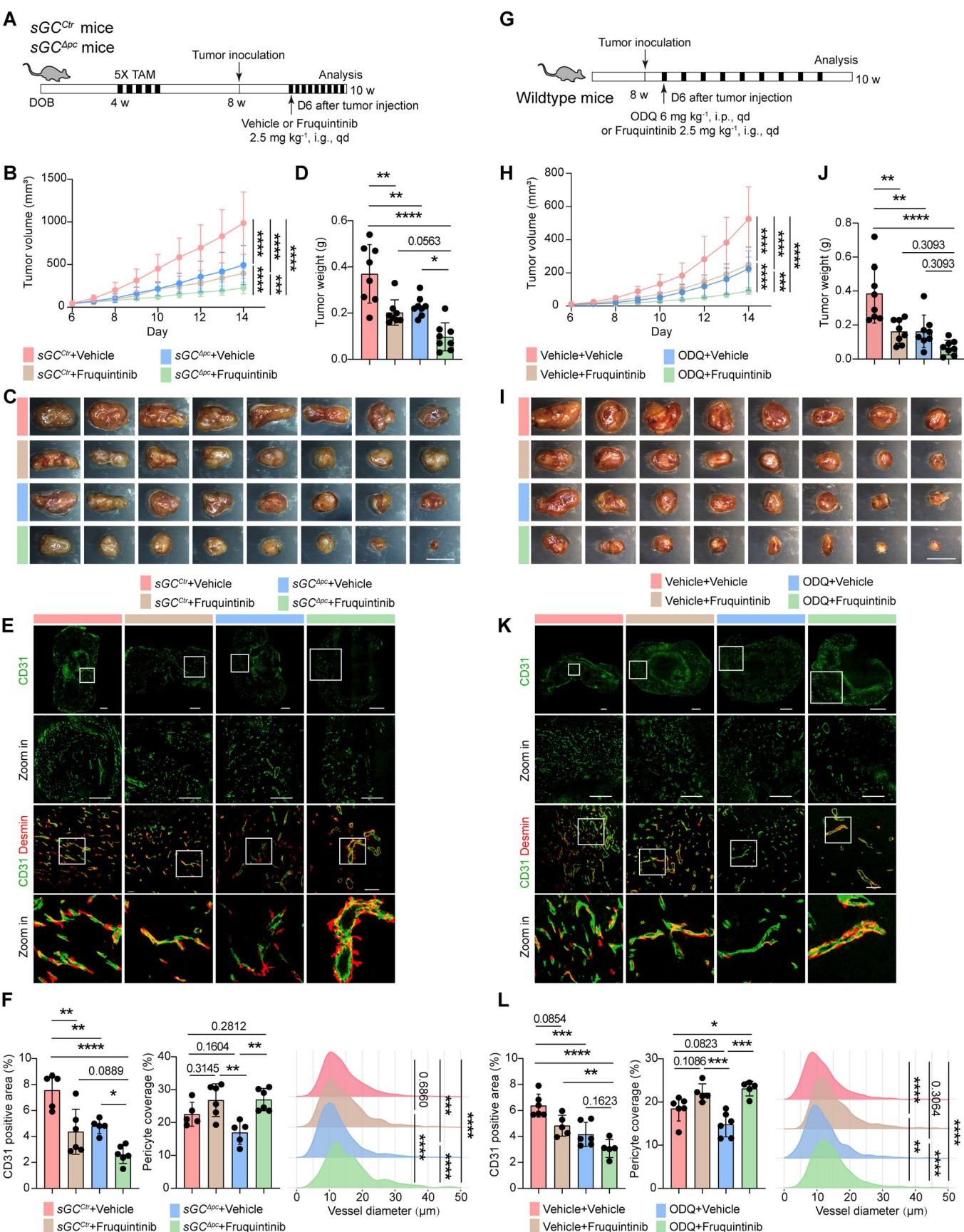

**Figure 6. Pericyte sGC disruption enhances anti-angiogenic therapy.**

(A) Schematic depiction of the experimental design. Both $sGC^{ctr}$ and $sGC^{\Delta PC}$ mice received five doses of tamoxifen at 4 weeks of age, followed by subcutaneous inoculation with LLC cells at 8 weeks of age. Starting from 6 post-tumor inoculation, mice were orally administered 2.5 mg kg$^{-1}$ Fruquintinib or corresponding vehicle daily, with tumor analysis scheduled at day 14 post-tumor injection. (B) Plot showing growth curves of LLC tumors in $sGC^{ctr}$ and $sGC^{\Delta PC}$ mice with Fruquintinib or vehicle treatment. Data presented as mean ± SD, with $n = 8$ mice per group. (C) Macroscope images of LLC tumors isolated from $sGC^{ctr}$ and $sGC^{\Delta PC}$ mice with Fruquintinib or vehicle treatment. Scale bars, 10 mm. (D) Plot showing weights of LLC tumors isolated from $sGC^{ctr}$ and $sGC^{\Delta PC}$ mice with Fruquintinib or vehicle treatment. Data presented as mean ± SD, with $n = 8$ mice per group. (E) Representative fluorescence images showing CD31 and Desmin-stained tumor sections from $sGC^{ctr}$ and $sGC^{\Delta PC}$ mice with Fruquintinib or vehicle treatment. Scale bars, 1 mm (CD31); 500 μm (CD31 Zoom-in); 100 μm (CD31 Desmin). (F) Plot showing the CD31-positive area, pericyte coverage, and vessel diameter in $sGC^{ctr}$ and $sGC^{\Delta PC}$ tumors with Fruquintinib or vehicle treatment. Data presented as mean ± SD, with $n = 5–6$ mice per group. (G) Schematic depiction of the experimental design. Wild-type mice were subcutaneously inoculated with LLC cells at 8 weeks of age, and then received oral administration of 2.5 mg kg$^{-1}$ Fruquintinib or corresponding vehicle and intraperitoneal injection of 6 mg kg$^{-1}$ ODQ or corresponding vehicle, both starting from day 6 post-tumor inoculation, with tumor analysis at day 14 post inoculation. (H) Plot depicting the growth curves of LLC tumors in wild-type mice with Fruquintinib or ODQ treatment. Data presented as mean ± SD, with $n = 8$ mice per group. (I) Macroscope images of LLC tumors isolated from wild-type mice with Fruquintinib or ODQ treatment. Scale bars, 10 mm. (J) Plot showing weights of LLC tumors isolated from wild-type mice with Fruquintinib or ODQ treatment. Data presented as mean ± SD, with $n = 8$ mice per group. (K) Representative fluorescence images showing CD31 and Desmin-stained tumor sections from wild-type mice with Fruquintinib or ODQ treatment. Scale bars, 1 mm (CD31); 500 μm (CD31 Zoom-in); 100 μm (CD31 Desmin). (L) Plot showing the CD31-positive area, pericyte coverage, and vessel diameter of wild-type mice with Fruquintinib or ODQ treatment. Data presented as mean ± SD, with $n = 5–6$ mice per group. Statistical significance assessed using two-way ANOVA test (B, H) or one-way ANOVA test with Tukey test. (D, F, J, L). *$P < 0.05$; **$P < 0.01$; ***$P < 0.001$; ****$P < 0.0001$. Source data are available online for this figure.

from $sGC^{\Delta PC}$ tumor or CD105$^{neg}$ CAFs (Figs. 5N,O and EV5G,H). These findings collectively demonstrated that pericytes and ECs within the vascular niche and their adjacent CAFs upregulated MIF expression following pericyte-specific sGC deletion, thereby synergistically skewing M2-like macrophages towards an M1-like phenotype, thereby contributing to the inhibition of tumor growth.

## Pericyte sGC disruption enhances anti-angiogenic therapy

Lastly, we examined whether impaired EC–pericyte crosstalk resulting from pericyte-specific sGC deletion would enhance the EC's sensitivity to anti-angiogenic therapy. To address this, we inoculated LLC cells into both $sGC^{Ctr}$ and $sGC^{\Delta PC}$ mice. Following a 6-day interval, the mice were administered either a vehicle or Fruquintinib continuously for 8 days. Fruquintinib, a multi-kinase inhibitor targeting VEGFR1/2/3, has received FDA approval for treating metastatic CRC and is presently undergoing phase-3 clinical trials for evaluating its efficacy against lung cancer (NCT02691299) (Dasari et al, 2023; Sun et al, 2014). As expected, Fruquintinib treatment significantly reduced tumor growth in $sGC^{Ctr}$ mice. Surprisingly, despite LLC tumors displaying smaller sizes in $sGC^{\Delta PC}$ mice compared to their $sGC^{Ctr}$ counterparts, Fruquintinib treatment further inhibited tumor growth in the $sGC^{\Delta PC}$ mice (Fig. 6A–D). Immunofluorescence analysis revealed that Fruquintinib reduced vessel density in both $sGC^{Ctr}$ and $sGC^{\Delta PC}$ LLC tumors. Notably, the number of CD31$^+$ vessels without pericyte coverage in $sGC^{\Delta PC}$ LLC tumors was significantly reduced by Fruquintinib treatment, resulting in a substantial increase in pericyte coverage. Interestingly, the remaining vessels in Fruquintinib-treated $sGC^{\Delta PC}$ LLC tumors exhibited a significantly larger diameter than vessels in the other groups. However, we observed that pericytes surrounding these remaining vessels in $sGC^{\Delta PC}$ LLC did not tightly ensheath the endothelium, as seen in $sGC^{Ctr}$ LLC tumors (Fig. 6E,F). This suggests that while Fruquintinib effectively targeted endothelial VEGFR signaling and inhibited tumor angiogenesis, it did not fully restore EC–pericyte interaction.

Given that pericyte sGC deletion did not affect healthy vessels (Appendix Fig. S1D,E), we further explored whether pharmacologically blocking NO-sGC signaling could recapitulate the vascular

defect observed with pericyte-specific sGC deletion and whether sGC blockade would enhance the efficacy of anti-angiogenic therapy. We first administered the WT adult mice with ODQ, a specific sGC inhibitor (Zhao et al, 2000), for 9 days, and we did not observe any notable change in vessel density, pericyte coverage, or albumin leakage (Appendix Fig. S9A–C). These results indicated that short-term ODQ treatment did not disrupt the integrity of a healthy vascular network. Then, we inoculated LLC tumors into 8-week-old WT mice and subsequently administered Fruquintinib and ODQ 6 days post-tumor inoculation. We observed that ODQ treatment alone reduced tumor growth, which was comparable to Fruquintinib treatment. Remarkably, the combination of ODQ and Fruquintinib further decreased tumor size (Fig. 6G–J). Consistent with these findings, the combination of ODQ and Fruquintinib treatment significantly impeded the growth of orthotopically inoculated EO771 (Appendix Fig. S10A–D). Histological analysis revealed that ODQ treatment markedly reduced vessel density and pericyte coverage. Particularly, the group receiving both ODQ and Fruquintinib displayed the lowest vessel density and larger vessel diameter, similar to the vessels observed in Fruquintinib-treated $sGC^{\Delta PC}$ LLC tumors (Fig. 6K,L). Interestingly, an analysis of The Cancer Genome Atlas datasets revealed that higher *GUCY1B1* expression is significantly associated with reduced overall survival in patients diagnosed with uveal melanoma, colorectal carcinoma, and mesothelioma (Appendix Fig. S11A), further substantiate the notion that suppressing sGC signaling holds therapeutic benefits. In summary, these results underscore the pivotal role of NO-sGC crosstalk in tumor vascularization and highlight the significant potential of disrupting NO-sGC crosstalk to sensitize tumors to anti-angiogenic therapy.

## Discussion

The role of blood vessels in supporting tumor growth via direct and indirect intercellular crosstalk, in addition to their conduit function, began to be recognized in the past decade. However, research has predominantly focused on the role of endothelial cells in regulating tumor microenvironment, and the specific contribution of pericytes has been understudied. In this study, we addressed

this gap by revealing the multifaceted role of pericytes in tumors. By deleting the NO receptor sGC in pericytes, we demonstrated that pericyte not only crosstalk to ECs to promote tumor vessel formation but also play an undiscovered role in shaping the intratumoral microenvironment by modulating CAFs and TAMs. Furthermore, our findings highlighted the significant enhancement in the efficacy of anti-angiogenic therapy through the disruption of endothelial cell-pericyte crosstalk, underscoring the therapeutic implications of our discoveries.

The role of eNOS-NO-sGC signaling in regulating vSMCs relaxation and vascular tone is widely recognized (Rees et al, 1989; Stasch et al, 2011). However, the expression of sGC in pericytes has only recently been identified (Bettaga et al, 2015; Friebe and Englert, 2022; Fukutani et al, 2009; He et al, 2023; He et al, 2016; Yang et al, 2021). In the current study, we have demonstrated the specific expression of sGC in tumor pericytes. It has been reported that ablation of endothelial cell-derived NO impaired pericyte recruitment in tumors (Kashiwagi et al, 2005). Interestingly, we observed that, in contrast to NO ablation, the deletion of its cognate receptor sGC in pericytes significantly disrupted the process of pericyte ensheathment around capillaries. Remarkably, this alteration did not significantly impact the recruitment of pericytes from neighboring tissues into the inoculated tumors. Hence, our findings suggest that pericyte sGC signaling is essential for establishing stable contacts between ECs and pericytes, facilitating the subsequent ensheathment process. This function is distinct from the role of PDGFB-PDGFRB signaling, which primarily governs pericyte recruitment to nascent endothelial sprouts.

The observed defects in pericyte behavior, including their inability to form stable contacts with ECs and their failure to envelop newly formed tumor vessels following pericyte-specific sGC deletion properly, may be attributed to increased cell motility. A prior study from our lab demonstrated that under both in vitro and in vivo conditions, the reduction of sGC expression in response to inflammatory challenges results in the retraction of pericyte cellular processes and their subsequent detachment from the endothelium. Whereas pharmacologically activating sGC successfully stabilizes pericytes and preserves pericyte coverage by dampening MRTFA/SRF activation and suppressing cytoskeleton reorganization in these cells (He et al, 2023). Interestingly, a separate study by Orlich et al illustrated that EC-secreted PDGFB activates MRTFA/SRF signaling to promote pericyte migration during angiogenesis (Orlich et al, 2022). These findings suggest an intricate interplay between PDGFB-PDGFRB signaling and NO-sGC signaling, which coordinates pericyte recruitment, migration, and ensheathment during angiogenesis through modulating MRTFA/SRF activity in pericytes.

The perturbation of sGC signaling within pericytes led to pronounced transcriptomic changes characterized by the down-regulation of genes associated with ECM deposition, encapsulation, and cell–matrix adhesion, as well as the upregulation of cytokines responsible for immune cell chemotaxis. Remarkably, the specific deletion of sGC in pericytes also triggered significant alterations in gene expression within ECs residing in the vascular niche. Specifically, genes linked to angiogenesis, such as *Tek*, *Kdr*, and *Pdgfb*, were significantly reduced. This finding aligns with previous studies highlighting the mutual influence of gene expression between pericytes and ECs. Interestingly, further analysis revealed that a substantial proportion of the altered genes in both ECs and

pericytes were not a direct outcome of sGC deletion. Rather, these changes stemmed from impaired Notch signaling, which requires direct cell–cell contact for activation. In addition, CellChat analysis of the intercellular communication also unveiled a dramatic disruption of Notch signaling between ECs and pericytes. This impairment was a consequence of the pericyte detachment triggered by sGC deletion. The profound impact of pericyte-specific gene deletion on adjacent endothelial cells within the vascular niche is also evident from an experiment in which the deletion of RBPj, a key transcriptional effector of Notch, within pericytes led to severe hypertrophy and degenerative changes in ECs, eventually resulting in the formation of cerebral cavernous malformation (CCM)-like lesions in the brain (Dieguez-Hurtado et al, 2019).

NO-sGC signaling is required for proper pericyte ensheathment within tumor vasculature, which undergoes continuous angiogenesis and remodeling. The administration of sGC inhibitor ODQ reduced pericyte coverage within tumors, paralleling the effect of pericyte-specific sGC deletion. Importantly, the combined treatment of Fruquintinib and ODQ exhibited a remarkable advantage over individual treatments. This aligns with previous studies that tumor vessels without pericyte coverage are more vulnerable to anti-angiogenic therapy (Benjamin et al, 1998; Bergers and Hanahan, 2008; Bergers et al, 2003). Importantly, our study also demonstrated that the role of sGC in quiescent pericytes might be dispensable, as the deletion of sGC in healthy mice did not compromise pericyte ensheathment and vascular integrity within the brain. This distinct outcome of sGC inactivation in angiogenic and quiescent pericytes provides a unique avenue for leveraging its therapeutic potential. By combining sGC inhibitors with anti-angiogenic agents, this approach could potentially yield enhanced therapeutic efficacy within tumor vasculature while sparing the vasculature in other organs from adverse effects.

Early investigations into the role of pericytes in tumors primarily relied on thymidine kinase-based pericyte ablation strategies (Cooke et al, 2012; Keskin et al, 2015). More recently, a study has demonstrated that pericytes, beyond their pivotal role in supporting tumor vessel formation, engage in direct crosstalk with tumor cells, exerting a significant influence on tumor growth. The data showed specific deletion of β3 integrin in pericytes does not affect tumor angiogenesis but instead fuels tumor growth by secreting CCL2, thereby directly activating the MEK1-ERK1/2-ROCK2 signaling pathway in tumor cells (Wong et al, 2020). While pericyte sGC deletion minimally altered the gene signature in tumor cells, it exerted a profound impact on CAFs and TAMs within the tumor microenvironment. Despite enhancing cellular motility and inducing transcriptomic reprogramming in pericytes, lineage tracing demonstrated that sGC deletion did not induce pericyte-to-fibroblast transition. Instead, the data support a model in which pericytes influence CAFs and TAMs through paracrine factors. Pericytes sGC inactivation resulted in a significant reduction of TGFβ1 expression in both ECs and pericytes, consequently driving the marked expansion of tumor-suppressive CD105$^{neg}$ CAFs. Notably, akin to pericytes in the atherosclerotic vessels (Pekayvaz et al, 2023), tumor pericytes upregulated MIF expression following sGC deletion, thereby skewing TAM polarization from the M2 to the M1 phenotype. These data underscore the important role played by pericyte in orchestrating tumor microenvironment and modulating tumor growth.

In conclusion, our study has provided valuable insights into the multifaceted role of pericytes in tumor biology. By targeting the NO receptor sGC in pericytes, we uncover their impact on tumor vascular niche and their modulation of CAFs and TAMs within the tumor microenvironment. Notably, inhibition of pericyte sGC presents an exciting avenue for augmenting the efficacy of anti-angiogenic therapy. These discoveries highlight the active contribution of pericytes in tumor development and underscore that a better understanding of the intricate interactions between pericytes, endothelial cells, and other cellular components in the tumor microenvironment would promote the development of novel therapeutic strategies and improve treatment outcomes.

# Methods

## Mice and genotyping

The generation of *Gucy1b1-flox* mice, and *Cspg4-CreER^{T2}* knock-in mice have been described previously (Huang et al, 2014; Zhang et al, 2011). *Gucy1a1-EGFP* transgenic mice (designated as *sGC-EGFP* mice), which express the EGFP under the control of sGC alpha subunit coding gene *Gucy1a1*, were acquired from GENSA-T.org. Wild-type C57BL/ 6J mice were purchased from Charles River (Beijing, China). All mice were housed within the specific-pathogen-free animal facility at the National Center for Protein Science. Genotyping primers are listed in Appendix Table S1. All animal experiments were performed according to the registered protocol (IACUC: 202003080011) that was approved by the Institutional Animal Care and Use Committee of the Interdisciplinary Research Center on Biology and Chemistry (IRCBC).

## Cell culture

LLC, B16F10, and Raw246.7 cells were cultured in DMEM medium (C11995500CP, Gibco™), supplemented with 10% FBS (FBS-12A, Capricorn), and 1% penicillin/streptomycin (15070063, Gibco™). EO771 cells were cultured in RPMI 1640 GlutaMAX medium (C11875500CP, Gibco™) supplemented with 10% FBS (FBS-12A, Capricorn), and 1% penicillin/streptomycin (15070063, Gibco™). Cells were cultured in a humidified atmosphere with 5% $CO_2$ at 37 °C. All cell lines underwent mycoplasma testing (Lonza, LT07318) prior to inoculation to ensure they were free from mycoplasma contamination. LLC, B16F10, and EO771 cells were gifts from Dr. Hellmut G. Augustin (Heidelberg University), Raw246.7 cell line was a gift from Dr. Lijian Hui (Shanghai Institute of Biochemistry and Cell Biology).

## In vivo tumor experiments

LLC ($1 \times 10^6$ cells), B16F10 ($1 \times 10^6$ cells), or EO771 ($1 \times 10^6$ cells) tumor cells were first suspended in 100 µl PBS, LLC and B16F10 tumor cells were subcutaneously injected into the right flank of sGC^{ΔPC} mice and their littermate controls, *sGC-EGFP* mice, or wild-type C57BL/6J mice, while EO771 tumor cells were orthotopically injected into the fat pad of fourth mammary gland in corresponding female mice. Tumor growth was monitored by measuring the length and width of tumors with calipers. The tumor volume was calculated using the formula: volume = length × width$^2$ × 0.52,

where length is the longest dimension and width is the corresponding perpendicular dimension. All mice were euthanized at the end of the experiments, then tumors were dissected, weighed, and photographed.

## Assessment of tumor vessel perfusion rate

To evaluate tumor vessel function, mice were intravenously injected with 100 µl saline containing FITC-conjugated Lectin-BSI (0.05 mg/mouse, L9381, Sigma) via the tail vein. Tumors were resected 15 min post-injection, and 10-µm-thick sections were prepared. The blood vessel perfusion rate was determined by quantifying lectin-positive vessels among CD31-stained vessels.

## In vivo Fruquintinib and ODQ treatment

Tumor-bearing *sGC^{ctr}* and *sGC^{ΔPC}* mice received administration of 2.5 mg kg$^{-1}$ Fruquintinib (S5667, Selleck) or corresponding vehicle by oral gavage daily from day 6 post-tumor injection, with tumor analysis scheduled at day 14. Tumor-bearing wild-type C57BL/6J mice received oral administration of 2.5 mg kg$^{-1}$ Fruquintinib or corresponding vehicle and intraperitoneal injection of 6 mg kg$^{-1}$ ODQ (HY-101255, MedChemExpress) or corresponding vehicle, both starting from day 6 post-tumor injection, and tumor analysis at day 14 post-tumor injection.

## Tumor dissection and dissociation

Tumors were gently minced into small pieces using scissors, and subsequently transferred into a 15-ml tube containing the digestion solution (2 mg ml$^{-1}$ Collagenase type IV (C5138, Sigma), 2.5 mg ml$^{-1}$ Dispase II (4942078001, Sigma), 40 µg ml$^{-1}$ DNase (10104159001, Sigma) in DMEM). Tumors were digested for 30 min at 37 °C with continuous rotation. Following digestion, the resulting cell suspensions were filtered through 70 µm cell strainers (CSS010070, Jetbiofil) placed on 50-ml tubes. The cell suspensions were subsequently centrifuged at $300 \times g$ for 7 min at 4 °C. The supernatants were aspirated completely. The cell pellets were resuspended in 5 ml 1× ACK buffer on ice for 3 min to lyse the erythrocytes. Next, 8 ml PBS was added into the tubes to stop reaction. In the end, the cells were centrifuged at $300 \times g$ for 7 min at 4 °C and resuspended with PBS for further analysis. Enrichment of CD146$^+$ cells was conducted using CD146 microbeads (130-092-007, Miltenyi Biotec) following the manufacturer's protocol.

## Single-cell RNA sequencing and bioinformatic analysis

Tumors were dissected 2 weeks after inoculation and subjected to enzymatic digestion. After dissociation, the viability (>85%) of tumor cells was determined using trypan blue staining. Single-cell suspensions from three tumors within the same group were pooled and loaded onto Chromium chips. Sequencing libraries were generated using the Chromium Single Cell 3' GEM Library & Gel Bead Kit v3 (PN-1000075,10X Genomics) according to the manufacturer's protocol. Sequencing data were processed with Seurat (v3) single-cell analysis pipeline with modifications. Raw fastq library was preprocessed by standard pipeline using Cell Ranger (v6.1.1) with mouse genome (mm10, GENCODE vM23/Ensembl 98) from 10X official website. Both generated raw count

matrices were loaded into R environment to check sequencing quality and filter with functions "perCellQCMetrics", "perFeatureQCMetrics" and "isOutlier" (with parameter "nmad" equal to 3) implemented in the scater (v1.22.0). Thereafter count matrix was converted to Seurat (v4) object with mitochondrial ratio QC (<10%) and median absolute deviation (MAD) feature QC (<3), followed by log-normalization. We selected 3000 highly variable genes for calculating PCA and integration using batchelor (v1.10.0) and first 20 diemension were used for clustering.

All clusters were manually annotated according to marker genes using function "FindAllMarkers" with Wilcoxon rank-sum test based on some of the well-known marker genes respectively (Pericytes: *Rgs5*, *Des* and *Pdgfrb*; Endothelial cells: *Pecam1*, *Cldn5*; Natural killer cells: *Cd74*; T cells: *Cd3d*; Cancer cell: *Krt18*; Fibroblasts: *Col1a1*, *Col1a2*; Granulocytes: *S100a9*; Macrophages: *Csf1r*), while "mixture" clusters with ambiguous (>1 cluster-specific markers for clusters defined above) markers were ruled out for clarity in terms of cell identity. At the end of filtering and processing, we retained overall 23,383 cells with 18,003 features (genes). Differentially expressed genes between $sGC^{ctr}$ and $sGC^{\Delta PC}$ mice were ranked based on the *P* values generated by method described above.

Fibroblasts and macrophages were re-clustered for finer resolution with distinct marker genes. For fibroblasts, we chose to remove cell cycle-related confounding effects using "CellCycleScoring" and "ScaleData" function from Seurat package as we observed high expression of *Cdk*-related genes and re-clustered them based on identified marker genes (*Ly6c1*, *CD74*, and *Acta2*). For macrophages, four clusters were re-clustering based on expression of *Il-1β*, *Ccr2*, *Adgre1*, *Mrc1*.

## Ligand receptors based cell–cell interaction analysis

Inference of cell–cell communication was done with CellChat (v1.6) separately for each library and then combined CellChat object from both libraries was used for comparison. All first-level clusters were used as cell-type labels for cluster-specific analysis and communications strength between ligands and receptors for pericytes and endothelial cells was based on absolute changed probability. Circos plots were produced from package circlize (v0.4.15).

## Velocity-based macrophage analysis

Macrophage trajectory analysis was conducted using python (v3.10) with scvelo (v0.2.5) package based on count matrices from velocyto (v0.17) output. We extracted previous-labeled macrophage cell from new-generated matrces and only retained common cells with previous output. Package fgsea (v1.20.0) was used to conduct GSEA analysis as well as plotting results.

## ChIP-seq data analysis

For ChIP-seq data, we grabbed data from GEO (project PRJNA150959). Burrows-Wheeler transformation (v0.7.17-r1188) was used for alignment with mouse genome mentioned above, followed by samtools (v1.17) filtering off reads with MAPQ smaller than 10. Peak calling for each sample (SRR397542 and SRR397543, with reference SRR397540 and SRR397541) was done by macs2

(v2.2.7.1) with additional option "—extsize 237" which derived from "mac2 predicted" option. Differential Peak from "macs2 bigdiff" was overlapped with "promoter area" defined by region covering upstream 2 kb and downstream 200 bp of each gene using bedtools (v2.31.0). "bamCoverage" command from deeptools (v3.5.1) with option "--normalizeUsing RPGC —binsize 20" and pyGenomeTracks (v3.8) were executed to generate peak coverage plots with simplified genome annotation and control data subtracted.

## Flow cytometry and cell sorting

After tissue-dissociation, cells were incubated with primary antibodies for 30 min at 4 °C, washed twice with PBS, and then resuspended in FACS buffer (1× PBS containing 0.5% FBS) (Cat. No. FBS-12A, Capricorn)). Cell surface Fc receptors were blocked with the anti-mouse CD16/CD32 mAb (1:200, 553141, BD Biosciences). Live/dead cells were discriminated using 7-AAD staining (0.25 µg/$1 \times 10^6$ cells, 559925, BD Biosciences). Cells were analyzed and sorted using the Beckman MoFlo Astrios.

To measure the percentage of endothelial cells and pericytes in LLC tumors, the following antibodies were used: FITC-conjugated anti-CD41 (1:200, 133903, Biolegend), FITC-conjugated anti-Ter119 (1:100, 557915, BD Biosciences), FITC-conjugated anti-CD45 (1:100, 553079, BD Biosciences), APC-conjugated anti-CD31 (1:100, 551262, BD Biosciences). CD41⁻Ter119⁻CD45⁻CD31⁺ cells were identified as endothelial cells. The following antibodies were used for quantifying pericytes in *sGC-EGFP* transgenic mice: APC-conjugated anti-CD41 (1:100, 133914, Biolegend), APC-Cy7-conjugated anti-Ter119 (1:200, 560509, BD Biosciences), AF700-conjugated anti-CD45 (1:100, 560510, BD Biosciences). CD41⁻Ter119⁻CD45⁻EGFP⁺ cells were identified as pericytes.

To analyze and purify CD105⁺ CAFs within LLC tumors, the following antibodies were used: PE-conjugated anti-Ter119 (1:100, 553673, BD Biosciences), PE-conjugated anti-CD45 (1:200, 561087, BD Biosciences), AF488-conjugated anti-PDPN (1:200, 53-5381, Thermo Fisher Scientific), APC-conjugated anti-CD105 (1:100, 561087, Biolegend). Ter119⁻CD45⁻PDPN⁺CD105⁺ cells and Ter119⁻CD45⁻PDPN⁺CD105⁻ cells were sorted as CD105ᵖᵒˢ CAFs and CD105ⁿᵉᵍ CAFs, respectively.

To analyze CD146⁺ cells in *Gucy1a1-EGFP* transgenic mice, the following antibodies were used: PE-conjugated anti-Ter119 (1:100, 553673, BD Biosciences), PE-conjugated anti-CD45 (1:200, 561087, BD Biosciences), PerCP-Vio700-conjugated anti-CD146 (1:10, 130-103-865, Miltenyi Biotec). Ter119⁻CD45⁻CD146⁺ cells were identified as CD146⁺ cells.

## Transwell assay

Assays were conducted using 12-mm transwells with 0.4-µm pore polyester membrane insert (3460, Corning). To investigate CD105ⁿᵉᵍ CAFs expansion promoted by CD146⁺ cells, CD146⁺ cells and CD105ᵖᵒˢ CAFs isolated from LLC tumors were resuspended in EC growth medium 2 (C-22011, PromoCell) supplemented with 1% penicillin/streptomycin (15070063, Gibco™). The upper chamber received $1 \times 10^6$ CD146⁺ cells, while the lower chamber contained $2 \times 10^5$ CD105ᵖᵒˢ CAFs. After 48 h, CAFs were harvested for measuring gene expression using qPCR.

For analyzing the impact of CD146⁺ cells on macrophage polarization, CD146⁺ cells isolated from LLC tumors and Raw246.7

cells were resuspended in EC growth medium 2 (C-22011, PromoCell) supplemented with 1% penicillin/streptomycin (15070063, Gibco™). In total, $1 \times 10^6$ CD146$^+$ cells were seeded into the upper chamber, while $2 \times 10^6$ Raw246.7 cells were seeded into the lower chamber. After 48 h, Raw246.7 cells were harvested, and the expression of M1 and M2-specific marker genes were measured using qPCR.

For investigating the impact of CAFs on macrophage polarization, CD105$^{pos}$ CAFs or CD105$^{neg}$ CAFs isolated from LLC tumors were co-cultured with Raw246.7 cells in fibroblast growth medium 2 (C-23020, PromoCell) supplemented with 1% Penicillin/Streptomycin (15070063, Gibco™). In brief, $5 \times 10^5$ CAFs were seeded into the upper chamber, while $2 \times 10^6$ Raw246.7 cells were seeded into the lower chamber. After 48 h, Raw246.7 cells were harvested, and the expression of M1 and M2-specific marker genes was measured using qPCR.

## ELISA analysis

To measure cytokines in tumor lysate, peritumor connective tissue and adipose tissue were carefully removed from tumors. Then tumors were quickly homogenized using the TissueLyser II (Qiagen) to ensure uniformity. Lysate was prepared using 1× PBS supplemented with a protease inhibitor cocktail (1:100, B14001; Sangon). following centrifugation, the supernatants were used for quantifying the concentration of MIF and active TGF-β1 with MIF ELISA kit (JL11079; Jianglaibio, China) and TGF-β1 ELISA kit (JL12223; Jianglaibio, China) according to the manufacturer's instructions.

To measure cytokines in cell supernatants, isolated CAFs and CD146$^+$ cells were cultured in fibroblast growth medium 2 (C-23020, PromoCell) supplemented with 1% penicillin/streptomycin (15070063, Gibco™) and in EC growth medium 2 (C-22011, PromoCell) supplemented with 1% penicillin/streptomycin (15070063, Gibco™), respectively. After 48 h, the supernatants of CAFs and CD146$^+$ cells were collected and subjected for ELISA analysis using MIF ELISA kit (JL11079; Jianglaibio, China) and active TGF-β1 ELISA kit (JL12223; Jianglaibio, China) according to the manufacturer's instructions.

## MIF treatment

Raw246.7 cells were subjected to 800 ng/ml mouse MIF (HY-P7388, MedChemExpress) for 48 h. Subsequently, Raw246.7 cells were harvested to assess macrophage polarization.

## ISO-1 treatment

Raw246.7 cells were subjected to ISO-1 treatment at a concentration of 100 μg/ml (S7732, Selleck) for 48 h. Subsequently, RNA of Raw246.7 cells were extracted for assessing the expression of genes associated with macrophage polarization.

## TGF-β1 treatment

To evaluate the impact of TGF-β1 treatment on *Eng* expression in CAFs, we isolated total CAFs and CD105$^{neg}$ CAFs from LLC tumors of wild-type C57BL/6J mice and cultured within fibroblast growth medium 2 (C-23020, PromoCell) supplemented with 1% penicillin/streptomycin (15070063, Gibco™). After cell attachment, CAFs were treated with 10 ng/ml Mouse TGF-β1 (7666-MB-005/CF, R&D Systems) for 48 h, Then CAFs were harvested for assessing *Eng* expression using qPCR.

## Immunofluorescent staining

Tumor samples were fixed with 4% PFA overnight at 4 °C, next, samples were immersed in 30% sucrose for 24 h, and then cryopreserved in optimal cutting temperature compound (OCT) at −80 °C. Sections (10 μm thickness) were cut at −20 °C with a cryostat (Leica), Cryosections were rinsed with PBS and subsequently permeabilized and immuno-blocked for 1 h at room temperature with 10% donkey serum (935; Absin), and 0.5% Triton-X100 in 1× PBS. Primary antibodies diluted in 10% donkey serum (935; Absin), 1% BSA (A600332; Sangon), 0.2% Tween-20 (A600560; Sangon) in 1× PBS, and then sections were incubated with them overnight at 4 °C. After three times washing with 0.3% Tween-20 (A600560; Sangon) in 1× PBS (30 min each time). Sections were incubated with secondary antibodies diluted in 10% donkey serum (935; Absin), 1% BSA (A600332; Sangon), 0.2% Tween-20 (A600560; Sangon) in 1× PBS for 1 h at room temperature. Following another three times washing, sections were mounted using ProLong mounting medium (P36961; Invitrogen). Following primary antibodies were used in this study: anti-Desmin antibody (1:200, Ab15200-1, Abcam), anti-CD31 antibody (1:200, MA3105, Thermo Fisher Scientific), anti-albumin Antibody (1:200, A90-134A, Thermo Fisher Scientific), anti-TER-119 antibody (1:100, 557915, BD Biosciences), Rabbit anti-GUCY1B1 antibody (1:100, ab154841, Abcam), Rabbit anti-pSMAD3 antibody (1:100, ab52903, Abcam), anti-PDGFRB antibody (1:100, 3169, Cell Signaling Technology), anti-CD146 antibody (1:10, 130-103-865, Miltenyi), anti-Collagen IV antibody (1:100, 2150-1470, Bio-Rad Laboratories), anti-NG2 antibody (1:200, Ab5320, Abcam), anti-CD45 antibody (1:200, 20103-1-AP, Proteintech), anti-NOTCH3 antibody (1:100, AF1308, R&D Systems), anti-eNOS antibody (1:200, 32027, Cell Signaling Technology), anti-DLL4 antibody (1:100, AF1389, R&D Systems), anti-Claudin-5 antibody (1:200, Ab15106, Abcam), anti-VEGFR2 antibody (1:200, 2479S, Cell Signaling Technology), anti-Tie2 antibody (1:200, AF762-SP, R&D Systems). Tumor hypoxia area was analyzed using the Hypoxyprobe™ Plus Kit (HP2-100Kit, Hypoxyprobe) according to the manufacturer's instruction.

## Western blot analysis

Lung tissues were weighted and lysed in ice-cold RIPA buffer (P0013B, Beyotime) supplemented with phosphatase inhibitor complex (C500017-0001, Sangon) and protease inhibitor cocktail (C600387-0001, Sangon) according to the manufacturer's instruction. After homogenization, samples were slowly rotated at 4 °C for 15 min for complete lysis. Then samples were centrifuged and the supernatants were collected. Protein concentrations were determined using the BCA assay (20201ES86, Yeasen). Proteins were separated using SDS polyacrylamide gels and transferred onto nitrocellulose membranes (10600001, Amersham Protran). The membranes were blocked with 5% BSA (A600332, Sangon) in TBS-T buffer (containing 0.1% Tween-20) for 1 h. Then incubated with primary antibodies at 4 °C overnight, and secondary antibody at room temperature for 1 h. The following primary antibodies were used in this study: Rabbit anti-β-ACTIN antibody (1:1000, 4967, Cell Signaling Technology), Rabbit anti-GUCY1A1 antibody

(1:500, G4280, Sigma), Rabbit anti-NOTCH3 antibody (1:500, ab23426, Abcam), Rabbit anti-GUCY1B1 antibody (1:1000, ab154841, Abcam). HRP-conjugated donkey anti-rabbit IgG (H + L) (1:10000, 711-066-152, Jackson ImmunoResearch) was used as secondary antibodies. Western blot was performed using Pierce™ ECL Western blotting substrate (32209, Thermo) according to the manufacturer's instruction.

## qPCR quantification

Total RNA was extracted using the Total RNA Extraction Kit (B511321-0100, Sangon), and reverse transcribed into cDNA using the TaKaRa Taq™ kit (R001A, Takara) according to the manufacturer's instruction. qPCR was performed using the PowerUp™ SYBR™ Green Master Mix (A25778, Applied Biosystems) on a QuantStudio™ 3 Real-Time PCR System (Applied Biosystems). Primer pairs used in this study are listed in Appendix Table S2.

## Statistics

Statistical analyses were performed using GraphPad Prism (v8) or R/Bioconductor packages. Student $t$ test was used for two-group comparisons, ANOVA with Tukey's Honest Significant Difference post hoc test was used for multiple-group comparisons. $P$ value < 0.05 was considered as statistical significance. All data are presented as mean ± SD.

# Data availability

All scRNA-seq data have been deposited in the Gene Expression Omnibus (GEO) database with accession No. GSE243313.

# Peer review information

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

## Acknowledgements

The authors would like to express our gratitude to Dr. Mahak Singhal for critical reading and suggestions. Special thanks are extended to Dr. Frank Kirchhoff and Dr. Wenhui Huang (University of Saarland, Germany) for generously providing the *Cspg4-CreER^T2* knock-in mice. The authors are also thankful to Dr. Zhengyu Li (Texas A&M Health Science Center, USA) for providing the *Gucy1b1^flox* mice. Our appreciation further extends to the dedicated staff members of the animal facility at IRCBC and the animal facility at the National Facility for Protein Science in Shanghai (NFPS), as well as the team of the Molecular Imaging System at Zhangjiang Laboratory for their excellent support. Synopsis figure was created with Biorender.com. This work was supported by the National Science Foundation of China (NFSC) (No. 31872797), the Shanghai Municipal Science and Technology Major Project (No. 2019SHZDZX02), and the Shanghai Key Laboratory of Aging Studies (No. 19DZ2260400).

## Author contributions

**Jing Zhu**: Conceptualization; Formal analysis; Validation; Investigation; Writing—original draft; Writing—review and editing. **Wu Yang**: Conceptualization; Formal analysis; Supervision; Validation; Investigation; Writing—original draft; Writing—review and editing. **Jianyun Ma**: Conceptualization; Data curation; Software; Formal analysis; Validation; Investigation; Visualization; Methodology; Writing—original draft; Writing—review and editing. **Hao He**: Conceptualization; Data curation; Software; Formal analysis; Validation; Investigation; Visualization; Methodology. **Zhen Liu**: Data curation; Software; Formal analysis; Validation; Investigation; Visualization; Methodology. **Xiaolan Zhu**: Formal analysis; Investigation; Methodology. **Xueyang He**: Formal analysis; Investigation; Methodology. **Jing He**: Formal analysis; Investigation; Methodology. **Zhan Chen**: Formal analysis; Investigation; Methodology. **Xiaoliang Jin**: Formal analysis; Investigation; Methodology. **Xiaohong Wang**: Conceptualization; Formal analysis; Supervision. **Kaiwen He**: Conceptualization; Formal analysis; Supervision. **Wu Wei**: Conceptualization; Formal analysis; Supervision; Writing—original draft; Writing—review and editing. **Junhao Hu**: Conceptualization; Resources; Data curation; Formal analysis; Supervision; Funding acquisition; Validation; Investigation; Visualization; Methodology; Writing—original draft; Project administration; Writing—review and editing.

## Disclosure and competing interests statement

The authors declare no competing interests.

# Expanded View Figures

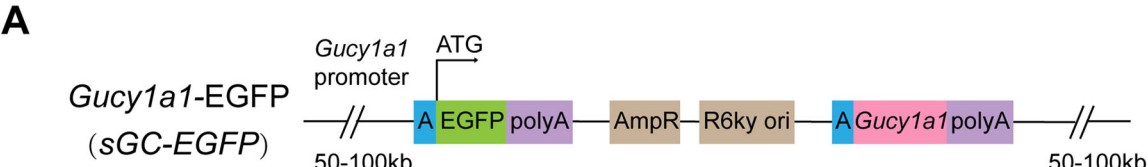

**A**

**B**

**C**

**D**

**Figure EV1.  sGC is specifically expressed in tumor vascular pericytes.**

(A) Schematic depiction of the construction strategy of *Gucy1a1-EGFP* (*sGC-EGFP*) transgenic mice. (B) Representative fluorescent images showing CD31, NG2, and Desmin-stained LLC tumor sections from *sGC-EGFP* mice. Scale bar, 50 μm. (C) Representative fluorescent images showing CD31, Desmin, and Flag-stained tumor sections from sGC-EGFP mice, demonstrating that tumor cells transduced with Flag-luciferase did not express sGC, as evidenced by the lack of co-localization with the sGC-EGFP signal. Scale bar, 50 μm. (D) Three-dimensional reconstruction depicting EGFP-expressing pericyte wrapping around a capillary within LLC tumors. Scale bar, 8 μm. Source data are available online for this figure.

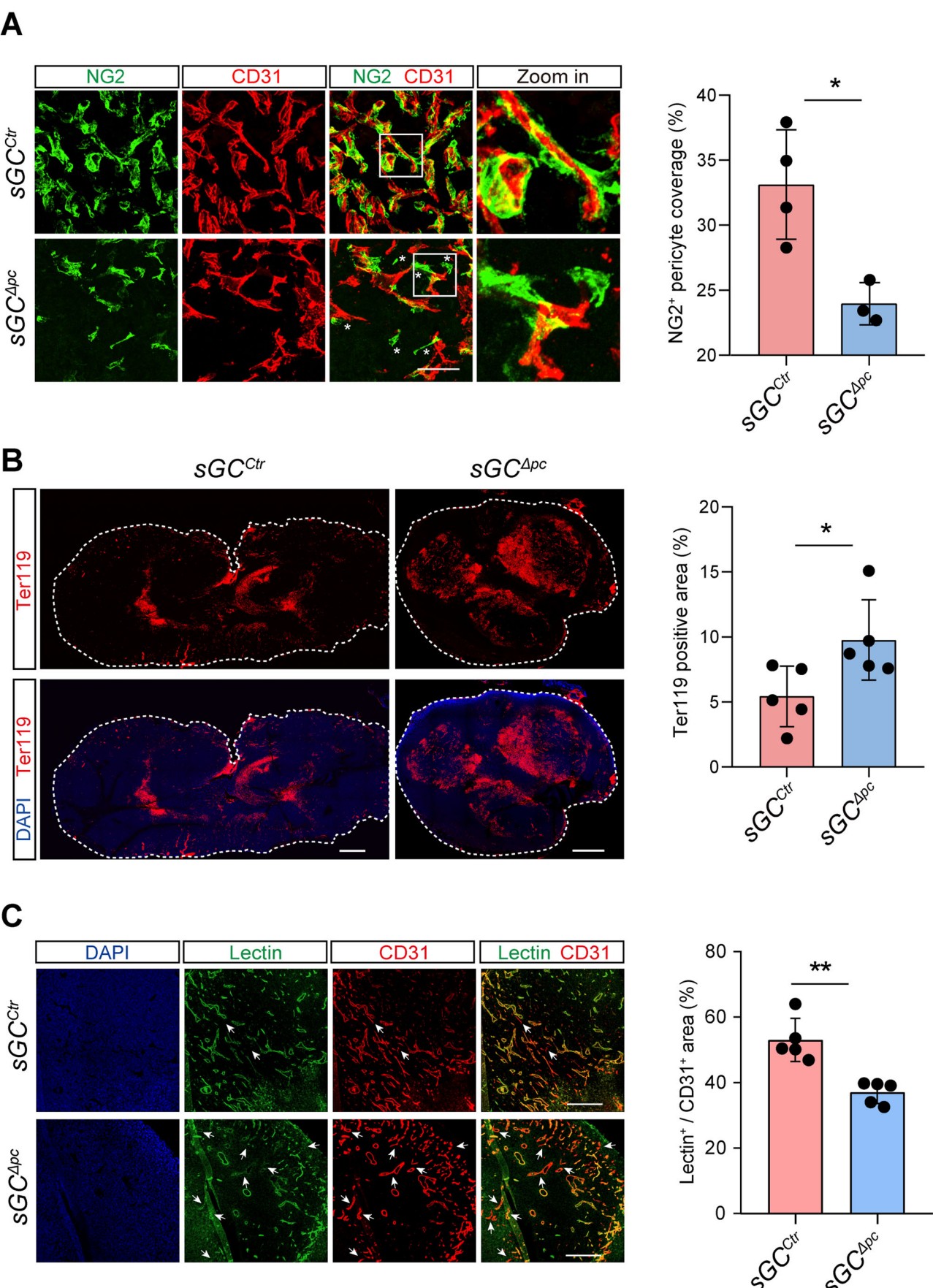

**Figure EV2.  Pericyte-specific sGC inactivation impairs blood vessel stability in LLC tumors.**

(A) Representative fluorescent images depicting CD31 and NG2-stained tumor sections from *sGC^{ctr}* and *sGC^{ΔPC}* mice. Plot showing the percentage of vessels covered by NG2-positive pericytes. Data presented as mean ± SD, with *n* = 3–4 mice per group. Scale bar, 100 μm. (B) Representative fluorescent images displaying DAPI and Ter119-stained LLC tumor sections from *sGC^{ctr}* and *sGC^{ΔPC}* mice. Plot showing the Ter119-positive area in tumors. Data presented as mean ± SD, with *n* = 5 mice per group. Scale bar, 1 mm. (C) Representative fluorescent images displaying DAPI, Lectin and CD31-stained LLC tumor sections obtained from *sGC^{ctr}* and *sGC^{ΔPC}* mice. Arrows indicate non-perfused vessels. Plot showing the functional vessel area in tumors. Data presented as mean ± SD, with *n* = 5 mice per group. Scale bar, 300 μm. Statistical significance assessed using two-tailed Student's *t* test (A–C). *P < 0.05; **P < 0.01. Source data are available online for this figure.

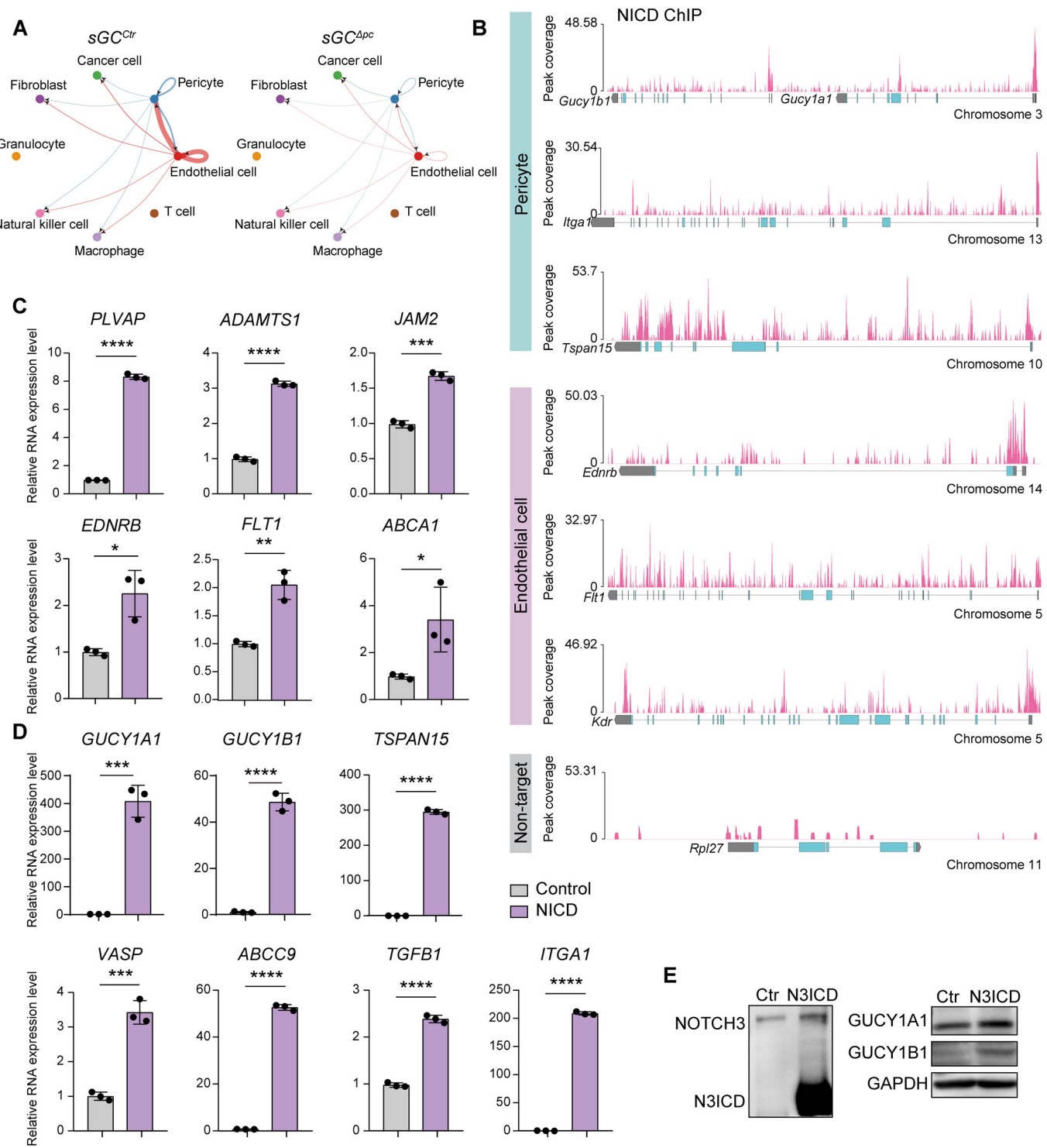

**Figure EV3. Pericyte-specific sGC inactivation alters Notch signaling pathway.**

(A) Circle plot showing the Notch signaling interactions among different cell populations within the tumors. Arrows indicate the direction of ligand-to-receptor interaction, while edge thickness reflects the cumulative weighted pathways between cell populations. (B) Analysis of NICD ChIP-seq dataset (GSE34954, comparing NICD-overexpressing cells to cells lacking Notch expression) reveals the enrichment of NICD at the promoter regions of *Gucy1a1, Gucy1b1, Itga1, Tspan15, Ednrb, Flt1, Kdr,* and *Rpl27.* The non-targeting gene *Rpl27* was included as a reference to confirm the specificity of NICD binding. (C, D) qPCR analysis of gene expression in control and NICD-overexpressing HUVECs (C) and HBVPs (D). Data presented as mean ± SD, with *n* = 3 replicates. (E) Western blot analysis of GUCY1A1 and GUCY1B1 protein levels in control and NICD-overexpressing HBVPs. Data presented as mean ± SD, with *n* = 3 replicates. Statistical significance assessed using two-tailed Student's *t* test (C, D). *$P < 0.05$; **$P < 0.01$; ***$P < 0.001$; ****$P < 0.0001$. Source data are available online for this figure.

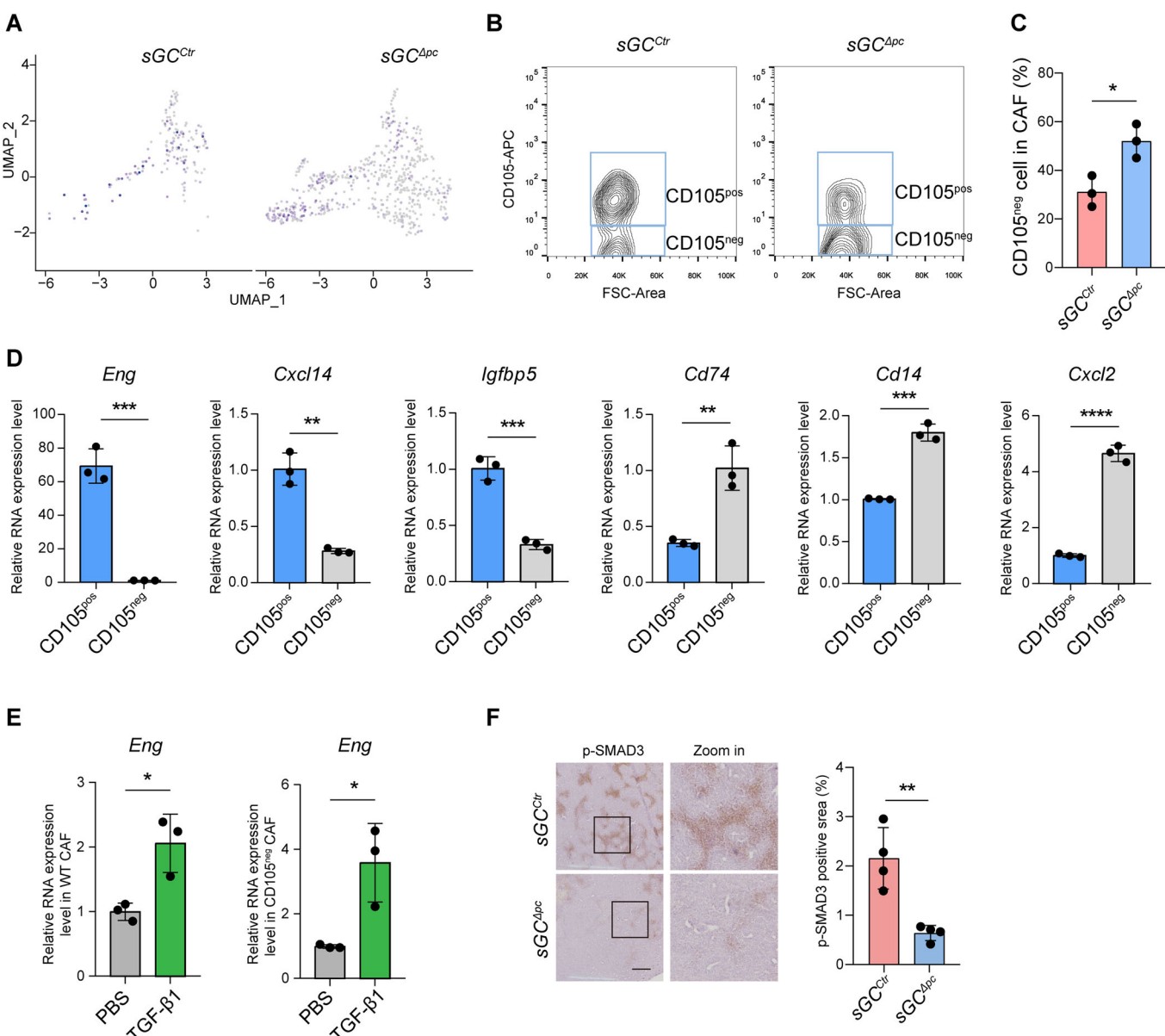

**Figure EV4. CD105^pos and CD105^neg CAFs identification.**

(A) UMAP plots showing *Eng* expression levels in CAFs. (B) FACS gating strategy employed for CD105^pos and CD105^neg CAFs isolation. (C) Plots showing the proportion of CD105^neg CAFs in *sGC^ctr* and *sGC^ΔPC* mice, Data presented as mean ± SD, with $n = 3$ mice per group. (D) qPCR analysis of the expression levels of *Eng, Cd74, Igfbp5, Cxcl14, Cd14*, and *Cxcl2* in CD105^pos and CD105^neg CAFs. CD105^pos and CD105^neg CAFs were isolated from five LLC tumors and pooled to ensure an adequate cell number for qPCR analysis. The plot depicts one representative dataset from two independent experimental repeats. Data presented as mean ± SD, with $n = 3$ technical replicates. (E) qPCR analysis of *Eng* expression in total CAFs and CD105^neg CAFs after TGF-β1 treatment. Data presented as mean ± SD. with n = 3 replicates. (F) Representative immunohistochemistry staining of pSMAD3 in *sGC^ctr* and *sGC^ΔPC* LLC tumors. The plot illustrates the quantification of the pSMAD3 positive area. Data presented as mean ± SD, with $n = 4$ mice per group. Scale bar, 500 μm. Statistical significance assessed using two-tailed Student's *t* test (C–F). *$P < 0.05$; **$P < 0.01$; ***$P < 0.001$; ****$P < 0.0001$. Source data are available online for this figure.

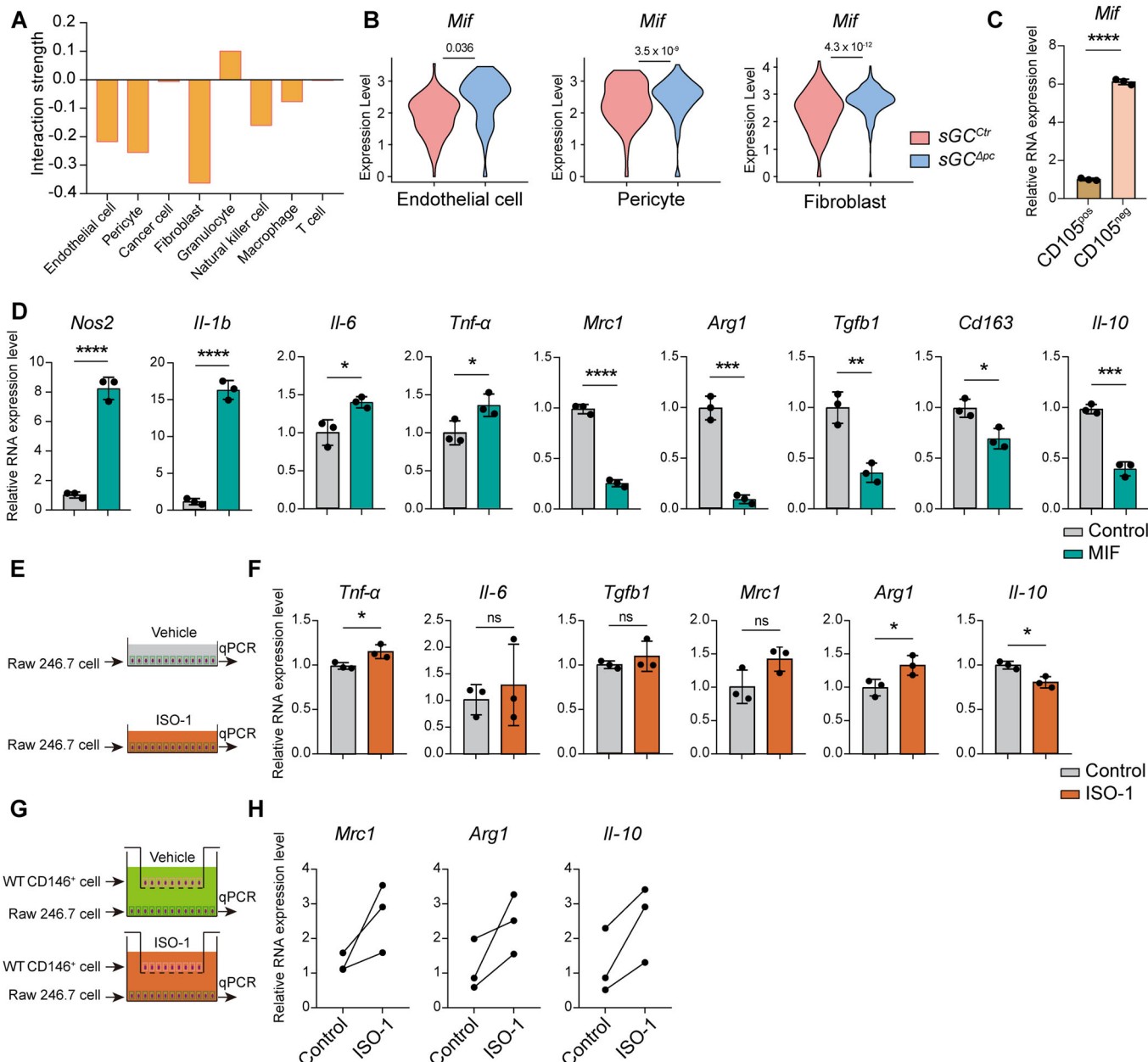

**Figure EV5. MIF promotes macrophage M1-like polarization.**

(A) The alteration in cell–cell interaction strength between TAMs and other cellular components in the tumors following pericyte-specific sGC inactivation. (B) Violin plots showing the expression levels of *Mif* in ECs, pericytes, and CAFs within *sGC^ctr* and *sGC^ΔPC* tumors. (C) Plot showing *Mif* expression levels in CD105^pos and CD105^neg CAFs. To obtain a sufficient number of cells, CD105^pos and CD105^neg CAFs were isolated from five LLC tumors and subsequently pooled for the experiment. Data presented as mean ± SD. (D) qPCR analysis of gene expression levels of *Nos2, Il-1b, Il-6, Tnf-a, Mrc1, Arg1, Tgfb1, Cd163,* and *Il-10* in Raw246.7 cells treated with either MIF or vehicle. Data presented as mean ± SD, with n = 3 replicates. (E) Schematic depiction of the experimental design. Raw246.7 cells were treated with either vehicle or ISO-1. (F) qPCR analysis of gene expression levels of *Il-6, Tnf-a, Mrc1, Arg1, Tgfb1,* and *Il-10* in Raw246.7 cells treated with ISO-1 or vehicle. Data presented as mean ± SD, with n = 3 replicates. (G) Schematic depiction of the experimental design. WT CD146+ cells were seeded into the transwell upper chamber, Raw246.7 cells in the bottom chamber. The culture medium was supplemented with either vehicle or ISO-1. (H) qPCR analysis of gene expression levels of *Mrc1, Arg1,* and *Il-10* in Raw246.7 cells treated with ISO-1 or vehicle. Data presented as mean ± SD, with n = 3 mice per group. Statistical significance assessed using unpaired two-samples Wilcoxon test (B), two-tailed Student's *t* test (C, D, F). *$P < 0.05$; **$P < 0.01$; ***$P < 0.001$; ****$P < 0.0001$. Source data are available online for this figure.

