## [Peer Review File · The EMBO Journal]

Pericyte signaling via soluble guanylate cyclase shapes the vascular niche and microenvironment of tumors

Junhao Hu, Jing Zhu, Wu Yang, Jianyun Ma, Hao He, Zhen Liu, Xiaolan Zhu, Xueyang He, Jing He, Zhan Chen, Xiaoliang Jin, Xiaohong Wang, Kai-Wen He, and Wu Wei

Corresponding author(s): Junhao Hu (jhu@sioc.ac.cn) , Wu Wei (wuwei@lglab.ac.cn)

Review Timeline:

Submission Date:	1st Nov 23
Editorial Decision:	29th Nov 23
Revision Received:	29th Jan 24
Editorial Decision:	29th Feb 24
Revision Received:	4th Mar 24
Accepted:	7th Mar 24

Editor: Ieva Gailite

Transaction Report:

Dear Dr. Hu,

Thank you for submitting your manuscript to The EMBO Journal. We have now received comments from two reviewers, which are included below for your information.

As you will see from the reports, the reviewers find the study interesting, while also indicating a number of concerns regarding the chosen tumour models and their analysis, as well as data analysis, presentation and interpretation. From the editorial side, I find the raised points generally reasonable. Therefore, I would like to invite you to address the comments of both reviewers in a revised version of the manuscript. I should add that it is The EMBO Journal policy to allow only a single major round of revision and that it is therefore important to resolve the main concerns at this stage. I think it would be useful to discuss the revision in more detail via email or phone/videoconferencing - please let me know which option you prefer.

We generally allow three months as standard revision time, which can be extended to six months in the case of major revisions. As a matter of policy, competing manuscripts published during this period will not negatively impact on our assessment of the conceptual advance presented by your study. However, please contact me as soon as possible upon publication of any related work to discuss the appropriate course of action. Should you foresee a problem in meeting this deadline, please let us know in advance to discuss an extension.

When preparing your letter of response to the referees' comments, please bear in mind that this will form part of the Review Process File and will therefore be available online to the community. For more details on our Transparent Editorial Process, please visit our website: <https://www.embopress.org/page/journal/14602075/authorguide#transparentprocess>. Please also see the attached instructions for further guidelines on preparation of the revised manuscript.

Please feel free to contact me if have any further questions regarding the revision. Thank you for the opportunity to consider your work for publication, and I look forward to receiving the revised manuscript.

Yours sincerely,

Ieva Gailite

Ieva Gailite, PhD
Senior Scientific Editor
The EMBO Journal
Meyerohofstrasse 1
D-69117 Heidelberg
Tel: +4962218891309
i.gailite@embojournal.org

- a point-by-point response to the referees' comments, with a detailed description of the changes made (as a word file).
- a word file of the manuscript text.
- individual production quality figure files (one file per figure)
- a complete author checklist, which you can download from our author guidelines (<https://www.embopress.org/page/journal/14602075/authorguide>).

- Expanded View files (replacing Supplementary Information)

We realize that it is difficult to revise to a specific deadline. In the interest of protecting the conceptual advance provided by the work, we recommend a revision within 3 months (27th Feb 2024). Please discuss the revision progress ahead of this time with the editor if you require more time to complete the revisions. Use the link below to submit your revision:

Referee #1:

Pericyte sGC signalling shapes tumor vascular niche and tumor microenvironment

Reviewers comments:

The paper provides an investigation into the role of pericytes, particularly focusing on the nitric oxide (NO) receptor soluble guanylate cyclase (sGC), in the tumor microenvironment. First, the study explores the impact of pericyte sGC deletion on the vascular niche and tumor growth. Next, utilizing single-cell RNA sequencing, the authors investigate the impact of pericyte sGC deletion on various TME populations, including cancer-associated fibroblasts (CAFs), tumor-associated macrophages (TAMs), pericytes, and endothelial cells, contributing to the overall depth of the study. The authors validate their findings in in vitro co-culturing experiments, reinforcing the observed effects. Notably, the paper concludes by suggesting a potential therapeutic strategy involving anti-angiogenic drugs, widening the study's scope and introducing a potential avenue for future research and clinical intervention in cancer treatment. The paper is well written and the narrative is logically constructed, which enhances its clarity and readability. Nevertheless, specific points for improvement and clarification have been identified and are elaborated upon in the subsequent sections.

Major concerns:

1. Tumor Models:

While the use of three different cell lines in in vivo experiments to investigate tumor growth is appreciated, subcutaneous tumors may not be the optimal model for studying the tumor microenvironment. Would it be possible to add at least one additional tumor model using an orthotopic model, such as mammary fat pad injections for breast cancer, to monitor tumor growth in the beginning of the story to enhance the clinical relevance of findings. It would also be good to use an orthotopic model to follow up on the treatment strategy in figure 6.

2. Clinical Correlation:

Consider exploring potential clinical correlations between pericyte sGC expression and tumor growth or patient survival. This could provide valuable insights into the clinical relevance of the study's findings.

3. Single-Cell Sequencing Analysis:

Clarify whether single-cell sequencing analysis was performed on size-matched tumors, as the deletion of sGC in pericytes appears to affect tumor size, potentially influencing cell composition. Also, what is the number of mice used for the single-cell sequencing?

4. Cspg4 Gene deletion as pericyte specific model:

Provide additional information on the choice of the Cspg4 gene for pericyte-specific knockout. Clarify why authors chose this gene and not another more common pericyte marker. Cspg4 is expressed in all smooth muscle cells so not really specific.

5. Data Representation:

Present all bar plots as simple bar plots with individual data points, as violin plots are typically used for larger datasets.

6. Statistical Analysis:

Add statistical analysis to Figures 4B, 4I, 4L, 4N, 5F-S13B for clarity.

Minor concerns:

Figure 1E-1H:

Increase the intensity of staining in Figures 1E and 1H for better visualization.

Figure 2E-2G:

Typo in title. What does 'upan' mean?

Include statistical analysis in Figure 2E-2G to demonstrate significant decreases in intercellular communication upon sGC deletion.

Figure 3B:

Can authors add fold change of the respective pathways in the figure in addition to p-value?

Figure 3H:

Authors need to provide additional explanation to clarify what they want to illustrate with this figure. This is not entirely clear from the text.

Figure 3I:

Include statistical information in Figure 3I to support the conclusion about reduced Notch ligands and receptor expression.

Supplementary Figure 1B-1C:

Add a tumor cell marker, such as pancytokeratin, to Figure S1B-1C to clearly demonstrate sGC expression in tumor-associated pericytes.

Supplementary Figure 2B:

Include quantification of western blot results. Consider providing additional evidence, such as IHC or IF stainings of the lung, to confirm pericyte-specific deletion of sGC.

Supplementary Figure 8B:

Could authors provide the ChIP-seq profile of the IgG control to confirm the enrichment of NICD binding to target genes? The current Chip profile appears somewhat noisy, and it's crucial to ascertain that this noise doesn't stem from nonspecific binding. Additionally, including the profile of a non-target gene would offer valuable insight into distinguishing specific binding patterns.

Supplementary Figure 8E:

Include WB quantification and provide an additional immunoblot to confirm NICD overexpression in this model.

Supplementary Figure 8CD:

Authors should provide bar plots with individual data points for clarity. The expression levels of GUCY1A1, GUCY1B1, and TSPAN15 appear markedly elevated, potentially out of proportion. Is there a specific reason for these substantial increases that deviate from the observed protein expression levels in figure 8E?

Supplementary Figure 9A:

Mention the upstream gating strategy for CAFs in the figure legend.

Supplementary Figure 11D:

The figure legend indicates that samples were pooled. Could authors specify whether these are technical or biological replicates in the qPCR analysis? Additionally, could authors provide information on the n-number of this experiment?

Supplementary Figure 13A:

Add a note in the figure legend specifying that this figure pertains to cell interactions concerning TAMs.

Referee #2:

Zhu et al present a study on the contribution of soluble guanylate cyclase (sGC) in pericytes to the tumor microenvironment (TME). The authors demonstrate that loss of sGC in pericytes does not impair vasculature in normal organs but results in modest reduction in tumor growth using three distinct murine tumor cell lines. Extensive scRNAseq analysis of tumors grown in WT or sGC mutant mice indicate that loss of sGC in pericytes alters the TME. The authors details changes in pericyte-EC interaction, TGF β , CAFs and macrophages. The authors then exploit this knowledge pharmacologically by inhibiting sGC with ODQ and demonstrate that ODQ enhances the efficacy of a multi-angioRTKi. Importantly the pharmacologic combination therapy provides more impressive anti-tumor activity then genetic loss of sGC alone. There are multiple interesting aspects of the study; however there are also some challenges as listed below.

Comments:

1. The vascular effects of the loss of sGC in pericytes are interesting.

- Inclusion of a tracer for functional vasculature would be an important addition to the characterization. Tomato-lectin or similar tools to mark functional vessels is recommended.

- The increased albumin and Ter119+ staining in tumors from sGC mutant mice suggests an increase in permeability of the tumor vasculature. Do the areas of elevated albumin correspond to elevated hypoxia? Can the authors provide additional explanation of what they believe is happening?

- the Hypoxyprobe data shown in Fig 1H indicates that the edge of the tumor is hypoxic but not the core, this is not typical. An overlay of CD31+ blood vessels with the hypoxyprobe is recommended. Additionally, does 1H shown the entire tumor, if so images showing increased magnification of the edge and central regions of the tumor should be shown.

2. The concept that loss of sGC in pericytes impacts TGF β activity in the tumor microenvironment is intriguing. However, there are a few things that should be clarified further.

- Figure 4M suggests via bioinformatics means that there is a reduction in TGF β activity in tumors in sGC mutant mice. This should be validated by looking at pSMAD2 or other measures of TGF β activity in tumors by IHC.

- Are the levels of TGF β quantified in Fig 4O, total or active TGF β , if total then active should be determined.

3. The authors suggest that the data in Fig 4 indicate that CD105 neg apCAFs contribute to the reduced tumor growth of tumors in sGC mutant mice. The data are correlative; there is no functional assessment of CD105 neg CAFs. I strongly recommend softening the language concluding this section. The contribution of apCAFs as tumor suppressive is tentative at best. The current best available data suggests that apCAFs can drive Treg formation in an antigen specific manner (PMID: 35523176) indicating that calling apCAFs tumor suppressive is challenging.

4. The data in Fig 5 is supportive of the loss of sGC in pericytes impacting myeloid cell phenotype. Please clarify if the CD105neg CAFs used in 5J,K are apCAFs?

Minor:

1. Supp fig 2 D, merge not shown; Supp fig 9B not all panels shown - I assume these are file transfer/image issues?

2. check the wording for the legend of Figure 3, in particular panel H. I believe the color scheme is referenced incorrectly.

3. Supp Fig 9, panel B, the graph y-axis is labeled VEGFR α positive area, this should be PDGFR α

4. A suggestion to increase impact. Perform IHC for adhesion molecules (ICAM, VCAM) and T cells in tumors treated with ODQ + Fruquintinib. The data in earlier parts of the manuscript suggest an immune-related mechanism and it is possible that the combo therapy enhances T cell infiltration.

1st Editorial Decision**November 29, 2023**

November 29, 2023

Manuscript EMBOJ-2023-116049 - Decision

Dear Dr. Hu,

Thank you for submitting your manuscript to The EMBO Journal. We have now received comments from two reviewers, which are included below for your information.

As you will see from the reports, the reviewers find the study interesting, while also indicating a number of concerns regarding the chosen tumour models and their analysis, as well as data analysis, presentation and interpretation. From the editorial side, I find the raised points generally reasonable. Therefore, I would like to invite you to address the comments of both reviewers in a revised version of the manuscript. I should add that it is The EMBO Journal policy to allow only a single major round of revision and that it is therefore important to resolve the main concerns at this stage. I think it would be useful to discuss the revision in more detail via email or phone/videoconferencing - please let me know which option you prefer.

We generally allow three months as standard revision time, which can be extended to six months in the case of major revisions. As a matter of policy, competing manuscripts published during this period will not negatively impact on our assessment of the conceptual advance presented by your study. However, please contact me as soon as possible upon publication of any related work to discuss the appropriate course of action. Should you foresee a problem in meeting this deadline, please let us know in advance to discuss an extension.

When preparing your letter of response to the referees' comments, please bear in mind that this will form part of the Review Process File and will therefore be available online to the community. For more details on our Transparent Editorial Process, please visit our website: <https://www.embopress.org/page/journal/14602075/authorguide#transparentprocess>. Please also see the attached instructions for further guidelines on preparation of the revised manuscript.

Please feel free to contact me if have any further questions regarding the revision. Thank you for the opportunity to consider your work for publication, and I look forward to receiving the revised manuscript.

Yours sincerely,

Ieva Gailite

Ieva Gailite, PhD

Senior Scientific Editor

The EMBO Journal

Meyerhofstrasse 1

D-69117 Heidelberg

Tel: +4962218891309

i.gailite@embojournal.org

See also guidelines for figure legends:

<https://www.embopress.org/page/journal/14602075/authorguide#figureformat>

Further information is available in our Guide For Authors:

We realize that it is difficult to revise to a specific deadline. In the interest of protecting the conceptual advance provided by the work, we recommend a revision within 3 months (27th Feb 2024). Please discuss the revision progress ahead of this time with the editor if you require more time to complete the revisions. Use the link below to submit your revision:

Referee #1:

Pericyte sGC signalling shapes tumor vascular niche and tumor microenvironment

Reviewers comments:

The paper provides an investigation into the role of pericytes, particularly focusing on the nitric oxide (NO) receptor soluble guanylate cyclase (sGC), in the tumor microenvironment. First, the

study explores the impact of pericyte sGC deletion on the vascular niche and tumor growth. Next, utilizing single-cell RNA sequencing, the authors investigate the impact of pericyte sGC deletion on various TME populations, including cancer-associated fibroblasts (CAFs), tumor-associated macrophages (TAMs), pericytes, and endothelial cells, contributing to the overall depth of the study. The authors validate their findings in in vitro co-culturing experiments, reinforcing the observed effects. Notably, the paper concludes by suggesting a potential therapeutic strategy involving anti-angiogenic drugs, widening the study's scope and introducing a potential avenue for future research and clinical intervention in cancer treatment. The paper is well written and the narrative is logically constructed, which enhances its clarity and readability. Nevertheless, specific points for improvement and clarification have been identified and are elaborated upon in the subsequent sections.

Major concerns:

1. Tumor Models:

While the use of three different cell lines in in vivo experiments to investigate tumor growth is appreciated, subcutaneous tumors may not be the optimal model for studying the tumor microenvironment. Would it be possible to add at least one additional tumor model using an orthotopic model, such as mammary fat pad injections for breast cancer, to monitor tumor growth in the beginning of the story to enhance the clinical relevance of findings. It would also be good to use an orthotopic model to follow up on the treatment strategy in figure 6.

2. Clinical Correlation:

Consider exploring potential clinical correlations between pericyte sGC expression and tumor growth or patient survival. This could provide valuable insights into the clinical relevance of the study's findings.

3. Single-Cell Sequencing Analysis:

Clarify whether single-cell sequencing analysis was performed on size-matched tumors, as the deletion of sGC in pericytes appears to affect tumor size, potentially influencing cell composition. Also, what is the number of mice used for the single-cell sequencing?

4. Cspg4 Gene deletion as pericyte specific model:

Provide additional information on the choice of the *Cspg4* gene for pericyte-specific knockout. Clarify why authors chose this gene and not another more common pericyte marker. *Cspg4* is expressed in all smooth muscle cells so not really specific.

5. Data Representation:

Present all bar plots as simple bar plots with individual data points, as violin plots are typically used for larger datasets.

6. Statistical Analysis:

Add statistical analysis to Figures 4B, 4I, 4L, 4N, 5F-S13B for clarity.

Minor concerns:

Figure 1E-1H:

Increase the intensity of staining in Figures 1E and 1H for better visualization.

Figure 2E-2G:

Typo in title. What does 'upan' mean?

Include statistical analysis in Figure 2E-2G to demonstrate significant decreases in intercellular communication upon sGC deletion.

Figure 3B:

Can authors add fold change of the respective pathways in the figure in addition to p-value?

Figure 3H:

Authors need to provide additional explanation to clarify what they want to illustrate with this figure.

This is not entirely clear from the text.

Figure 3I:

Include statistical information in Figure 3I to support the conclusion about reduced Notch ligands and receptor expression.

Supplementary Figure 1B-1C:

Add a tumor cell marker, such as pancytokeratin, to Figure S1B-1C to clearly demonstrate sGC expression in tumor-associated pericytes.

Supplementary Figure 2B:

Include quantification of western blot results. Consider providing additional evidence, such as IHC or IF stainings of the lung, to confirm pericyte-specific deletion of sGC.

Supplementary Figure 8B:

Could authors provide the ChIP-seq profile of the IgG control to confirm the enrichment of NICD binding to target genes? The current Chip profile appears somewhat noisy, and it's crucial to ascertain that this noise doesn't stem from nonspecific binding. Additionally, including the profile of a non-target gene would offer valuable insight into distinguishing specific binding patterns.

Supplementary Figure 8E:

Include WB quantification and provide an additional immunoblot to confirm NICD overexpression in this model.

Supplementary Figure 8CD:

Authors should provide bar plots with individual data points for clarity. The expression levels of GUCY1A1, GUCY1B1, and TSPAN15 appear markedly elevated, potentially out of proportion. Is there a specific reason for these substantial increases that deviate from the observed protein expression levels in figure 8E?

Supplementary Figure 9A:

Mention the upstream gating strategy for CAFs in the figure legend.

Supplementary Figure 11D:

The figure legend indicates that samples were pooled. Could authors specify whether these are technical or biological replicates in the qPCR analysis? Additionally, could authors provide information on the n-number of this experiment?

Supplementary Figure 13A:

Add a note in the figure legend specifying that this figure pertains to cell interactions concerning TAMs.

Referee #2:

Zhu et al present a study on the contribution of soluble guanylate cyclase (sGC) in pericytes to the tumor microenvironment (TME). The authors demonstrate that loss of sGC in pericytes does not impair vasculature in normal organs but results in modest reduction in tumor growth using three distinct murine tumor cell lines. Extensive scRNAseq analysis of tumors grown in WT or sGC mutant mice indicate that loss of sGC in pericytes alters the TME. The authors details changes in pericyte-EC interaction, TGF β , CAFs and macrophages. The authors then exploit this knowledge pharmacologically by inhibiting sGC with ODQ and demonstrate that ODQ enhances the efficacy of a multi-angioRTKi. Importantly the pharmacologic combination therapy provides more impressive anti-tumor activity than genetic loss of sGC alone. There are multiple interesting aspects of the study; however there are also some challenges as listed below.

Comments:

1. The vascular effects of the loss of sGC in pericytes are interesting.

- Inclusion of a tracer for functional vasculature would be an important addition to the characterization. Tomato-lectin or similar tools to mark functional vessels is recommended.

- The increased albumin and Ter119+ staining in tumors from sGC mutant mice suggests an increase in permeability of the tumor vasculature. Do the areas of elevated albumin correspond to elevated hypoxia? Can the authors provide additional explanation of what they believe is happening?

- the Hypoxyprobe data shown in Fig 1H indicates that the edge of the tumor is hypoxic but not the core, this is not typical. An overlay of CD31+ blood vessels with the hypoxyprobe is recommended. Additionally, does 1H shown the entire tumor, if so images showing increased magnification of the edge and central regions of the tumor should be shown.

2. The concept that loss of sGC in pericytes impacts TGF β activity in the tumor microenvironment is intriguing. However, there are a few things that should be clarified further.

- Figure 4M suggests via bioinformatics means that there is a reduction in TGF β activity in tumors in sGC mutant mice. This should be validated by looking at pSMAD2 or other measures of TGF β activity in tumors by IHC.

- Are the levels of TGF β quantified in Fig 4O, total or active TGF β , if total then active should be determined.

3. The authors suggest that the data in Fig 4 indicate that CD105 neg apCAFs contribute to the reduced tumor growth of tumors in sGC mutant mice. The data are correlative; there is no functional assessment of CD105 neg CAFs. I strongly recommend softening the language concluding this section. The contribution of apCAFs as tumor suppressive is tentative at best. The current best available data suggests that apCAFs can drive Treg formation in an antigen specific manner (PMID: 35523176) indicating that calling apCAFs tumor suppressive is challenging.

4. The data in Fig 5 is supportive of the loss of sGC in pericytes impacting myeloid cell phenotype. Please clarify if the CD105neg CAFs used in 5J,K are apCAFs?

Minor:

1. Supp fig 2 D, merge not shown; Supp fig 9B not all panels shown - I assume these are file transfer/image issues?
2. check the wording for the legend of Figure 3, in particular panel H. I believe the color scheme is referenced incorrectly.
3. Supp Fig 9, panel B, the graph y-axis is labeled VEGFR α positive area, this should be PDGFR α
4. A suggestion to increase impact. Perform IHC for adhesion molecules (ICAM, VCAM) and T cells in tumors treated with ODQ + Fruquintinib. The data in earlier parts of the manuscript suggest an immune-related mechanism and it is possible that the combo therapy enhances T cell infiltration.

Referee #1

Pericyte sGC signalling shapes tumor vascular niche and tumor microenvironment

Reviewers comments:

The paper provides an investigation into the role of pericytes, particularly focusing on the nitric oxide (NO) receptor soluble guanylate cyclase (sGC), in the tumor microenvironment. First, the study explores the impact of pericyte sGC deletion on the vascular niche and tumor growth. Next, utilizing single-cell RNA sequencing, the authors investigate the impact of pericyte sGC deletion on various TME populations, including cancer-associated fibroblasts (CAFs), tumor-associated macrophages (TAMs), pericytes, and endothelial cells, contributing to the overall depth of the study. The authors validate their findings in in vitro co-culturing experiments, reinforcing the observed effects. Notably, the paper concludes by suggesting a potential therapeutic strategy involving anti-angiogenic drugs, widening the study's scope and introducing a potential avenue for future research and clinical intervention in cancer treatment. The paper is well written and the narrative is logically constructed, which enhances its clarity and readability. Nevertheless, specific points for improvement and clarification have been identified and are elaborated upon in the subsequent sections.

Major concerns:

1. Tumor Models:

While the use of three different cell lines in in vivo experiments to investigate tumor growth is appreciated, subcutaneous tumors may not be the optimal model for studying the tumor microenvironment. Would it be possible to add at least one additional tumor model using an orthotopic model, such as mammary fat pad injections for breast cancer, to monitor tumor growth in the beginning of the story to enhance the clinical relevance of findings. It would also be good to use an orthotopic model to follow up on the treatment strategy in figure 6.

Response: We appreciate the reviewer's valuable suggestion. To further validate the results observed in sGC^{Ctr} and $sGC^{\Delta PC}$ mice, as well as in the settings with Fruquinitinib

and ODQ treatment in WT mice in a more physiological condition, we orthotopically injected EO771 breast cancer cells into the mammary fat pad. The analysis revealed that orthotopic EO771 tumors were significantly smaller in $sGC^{\Delta PC}$ mice than those in sGC^{Ctr} mice, faithfully recapitulating the phenotype observed in the subcutaneous model. The new data has been incorporated into the revised manuscript, replacing the data of subcutaneous model (**Appendix Figure S2D-F**). Furthermore, consistent with the results from the subcutaneously inoculated LLC tumor model, ODQ strongly enhanced the tumor-suppressing efficacy of Fruquinitinib treatment in orthotopically inoculated EO771 tumors in WT mice. This new data has now been included in the revised manuscript as **Appendix Figure S10A-D**.

2. Clinical Correlation:

Consider exploring potential clinical correlations between pericyte sGC expression and tumor growth or patient survival. This could provide valuable insights into the clinical relevance of the study's findings.

Response: We appreciate the reviewer for the constructive suggestion. Analysis of TCGA data revealed that higher expression levels of sGC in tumors are associated with reduced survival in uveal melanoma, colorectal adenocarcinoma, and mesothelioma (**Appendix Figure S11**).

3. Single-Cell Sequencing Analysis:

Clarify whether single-cell sequencing analysis was performed on size-matched tumors, as the deletion of sGC in pericytes appears to affect tumor size, potentially influencing cell composition. Also, what is the number of mice used for the single-cell sequencing?

Response: The scRNA-seq analysis was conducted two weeks post-tumor inoculation; therefore, the tumors analyzed were not size-matched at that time point. This specific information has been incorporated into the revised methods section for clarity.

Following the reviewer's suggestion, we performed an additional experiment and collected size-matched LLC tumors from both sGC^{Ctr} and $sGC^{\Delta PC}$ mice. We examined the proportions of ECs, pericytes, and CAFs using staining. The results demonstrated that

the alteration in the proportions of these major cell types between size-matched tumors of sGC^{Ctr} and $sGC^{\Delta PC}$ mice were comparable to the results obtained from non-size-matched tumors of the two genotypes. This observation affirms that the observed differences in cell composition are attributed to pericyte-specific sGC deletion rather than variations in tumor size (**Rebuttal Figure 1**).

In terms of the number of mice included in the scRNA-seq analysis, we collected tumors from three sGC^{Ctr} mice and three $sGC^{\Delta PC}$ mice. This information is included in the methods section of the supplementary materials.

4. *Cspg4* Gene deletion as pericyte specific model:

Provide additional information on the choice of the *Cspg4* gene for pericyte-specific knockout. Clarify why authors chose this gene and not another more common pericyte marker. *Cspg4* is expressed in all smooth muscle cells so not really specific.

Response: We appreciate the reviewer's insightful comment regarding the choice of the *Cspg4-CreER^{T2}* for pericyte-specific knockout and acknowledge that *Cspg4*, which encodes NG2, is not exclusively expressed in pericytes. It is indeed true that both *Cspg4* and *Pdgfrb*, the two most widely used pericyte markers, are expressed in pericytes as well as vascular smooth muscle cells (vSMCs). *Cspg4-Cre/CreER^{T2}* has been widely employed for deleting genes in pericytes of various organs and tumors and showed

comparable specificity and efficiency to *Pdgfrb-Cre/CreER^{T2}* (Hosaka, 2016; Mayr, 2022; Sato, 2016; Teichert, 2017). Therefore, we believe that the findings and conclusions of the current study will not be affected by the choice between *Cspg4-CreER^{T2}* or *Pdgfrb-CreER^{T2}*.

To address concerns regarding potential contributions from vSMCs to the vascular niche and tumor microenvironment following *Cspg4-CreER^{T2}*-driven sGC deletion, we conducted additional analyses. LLC tumors from *Cspg4-CreER^{T2}::Rosa26-LSL-tdTomato*

mice were stained with antibodies against CD31 and Calponin, a specific vSMC marker (Vanlandewijck, 2018). The analysis revealed that the majority of vessels were covered by tdTomato-positive pericytes, with only a few vessels wrapped by tdTomato- and Calponin-double positive vSMCs (**Rebuttal Figure 2A-B**). This observation demonstrated that the predominant cell population labeled by *Cspg4-CreERT²* in LLC tumors is pericytes. Additionally, scRNA-seq data analysis showed almost negligible expression of the Calponin-encoding gene *Cnn1* in all captured cell types, including *Cspg4*- and *Pdgfrb*-expressing pericytes (**Rebuttal Figure 2C**). This finding indicates a low cell number of vSMC in the tumors. These data conclusively suggest that changes in the vascular niche and tumor microenvironment following sGC deletion are primarily attributed to pericytes.

5. Data Representation:

Present all bar plots as simple bar plots with individual data points, as violin plots are typically used for larger datasets.

Response: We have updated the violin plots to bar plots with individual data points in the revised manuscript, as per the reviewer's recommendation.

6. Statistical Analysis:

Add statistical analysis to Figures 4B, 4I, 4L, 4N, 5F-S13B for clarity.

Response: We have added statistical analysis to **Figure 4N** and **Figure EV5B** (previously Supplementary Figure 13B) in the revised manuscript.

Figures 4B, 4I, 4K, and 5F compared the cell numbers or proportions of specific cell clusters between *sGC^{Ctrl}* and *sGC^{ΔPC}* LLC tumors. It is noteworthy that, as part of our experimental design to reduce the cost, we pooled cells isolated from three different tumors of the same group and loaded on to a single Chromium chip. While this approach reduced experimental costs, we now recognize that it introduced a limitation by decreasing the effective sample size (N) from three to one, therefore making it infeasible to perform statistical analysis on some of the data, such as **Figures 4B, 4I, 4K, and 5F**.

In contrast, in the case of **Figure 4N** and **Figure EV5B** and other figure that compared the gene expression levels between sGC^{Ctr} and $sGC^{\Delta PC}$ LLC tumors, the effective N corresponds to the cell number of a specific cluster. Thus, statistical analysis can be successfully performed in these instances.

Minor concerns:

Figure 1E-1H:

Increase the intensity of staining in Figures 1E and 1H for better visualization.

Response: The intensity of staining in **Figure 1E** and **1H** have been increased.

Figure 2E-2G:

Typo in title. What does 'upan' mean?

Include statistical analysis in Figure 2E-2G to demonstrate significant decreases in intercellular communication upon sGC deletion.

Response: We apologize for the mistake. The typo in the title of **Figure 2** has been corrected. Regarding the absence of statistical analysis in **Figure 2E-2G**, as explained in **Response-6**, this limitation persists due to the constraints outlined earlier. We acknowledge and thank the reviewer for their understanding in this matter.

Figure 3B:

Can authors add fold change of the respective pathways in the figure in addition to p-value?

Response: The pathway enrichment analysis is conducted based on the differential expression of genes (DEGs). Given the nature of this analysis, it focuses on the identification of statistically significant changes rather than fold changes. Therefore, the addition of fold change information to respective pathways is not feasible.

Figure 3H:

Authors need to provide additional explanation to clarify what they want to illustrate with this figure. This is not entirely clear from the text.

Response: We appreciate the reviewer's feedback. The **Figure 3H** legend has been rephrased for improved clarity.

Figure 3I:

Include statistical information in Figure 3I to support the conclusion about reduced Notch ligands and receptor expression.

Response: The statistical information has been added to **Figure 3I** of the revised manuscript.

Supplementary Figure 1B-1C:

Add a tumor cell marker, such as pancytokeratin, to Figure S1B-1C to clearly demonstrate sGC expression in tumor-associated pericytes.

Response: We appreciate the reviewer's insightful suggestion. Unfortunately, our attempts at pancytokeratin staining with tumors were not successful. Instead, since the tumor cells were transduced with Flag-luciferase, we stained the tumor cells with an antibody against the Flag tag. The obtained results clearly indicate that sGC expression is confined to pericytes within the tumor (**Figure EV1C**). Additionally, our single-cell analysis revealed that the expression of *Gucy1a1* and *Gucy1b1* was specifically restricted to the pericyte population. Collectively, these findings conclusively demonstrate that sGC expression is specifically expressed in tumor-associated pericytes (**Rebuttal Figure 3**).

Supplementary Figure 2B:

Include quantification of western blot results. Consider providing additional evidence, such as IHC or IF stainings of the lung, to confirm pericyte-specific deletion of sGC.

Response: Following the reviewer's suggestion, we have quantified the WB bands and performed additional IF staining to validate the pericyte-specific deletion of sGC in the lung. These data have been incorporated into **Appendix Figure S1B-C** of the revised manuscript.

Supplementary Figure 8B:

Could authors provide the ChIP-seq profile of the IgG control to confirm the enrichment of NICD binding to target genes? The current Chip profile appears somewhat noisy, and it's crucial to ascertain that this noise doesn't stem from nonspecific binding. Additionally, including the profile of a non-target gene would offer valuable insight into distinguishing specific binding patterns.

Response: We appreciate the reviewer for bringing the issue to our attention. We sincerely apologize for the inadvertent overlay of NICD ChIP-seq peaks (depicted in red) with peaks derived from ChIP-seq input reads (shown in gray).

The original dataset (GSE34954) identified NICD-bound loci by comparing NICD antibody-enriched peaks between NICD-overexpressing cells with control cells lacking Notch expression, ensuring the specificity of NICD binding. The original study (PMID:22237151) mentioned that ChIP-seq with IgG control was performed; however, no obvious peaks were detected in any of the samples. Therefore, ChIP-seq data for IgG control were not upload to the GEO. We have now corrected the figure to accurately represent the NICD binding profile. Additionally, a more detailed description of the experiment has been incorporated into the corresponding figure legend in **Figure EV3B** of the revised manuscript.

Supplementary Figure 8E:

Include WB quantification and provide an additional immunoblot to confirm NICD overexpression in this model.

Response: Following the reviewer's suggestion, we re-performed the NICD overexpression in pericytes and validated both the overexpression of NICD and the protein levels of downstream GUCY1A1 and GUCY1B1 using Western blotting. This data is now presented in **Figure EV3E** of the revised manuscript.

Supplementary Figure 8CD:

Authors should provide bar plots with individual data points for clarity. The expression levels of GUCY1A1, GUCY1B1, and TSPAN15 appear markedly elevated, potentially out of proportion. Is there a specific reason for these substantial increases that deviate from the observed protein expression levels in figure 8E?

Response: Individual data points have been incorporated into the bar plots as per the reviewer's request. Regarding the significant elevation of mRNA levels following NICD overexpression in HBVP without a proportional increase in protein levels, this discrepancy may be attributed to complex regulatory mechanisms, including mRNA stability, translational efficiency, and protein degradation. These factors can contribute to the decoupling of mRNA and protein levels.

Supplementary Figure 9A:

Mention the upstream gating strategy for CAFs in the figure legend.

Response: Detailed information about the upstream gating strategy for CAFs has been added to the figure legend (**Appendix Figure S6A** of the revised manuscript).

Supplementary Figure 11D:

The figure legend indicates that samples were pooled. Could authors specify whether these are technical or biological replicates in the qPCR analysis? Additionally, could authors provide information on the n-number of this experiment?

Response: Due to the challenges in isolating a sufficient number of CD105^{pos} and CD105^{neg} CAFs, cells isolated from five tumors were combined to form one sample for qPCR analysis. The experiments were independently conducted twice, yielding

consistent results, and only one representative set of data is shown in the figure. The data points in the plot represent three technical replicates. We have now updated the information in the figure legend. (**Figure EV4D** of the revised manuscript)

Supplementary Figure 13A:

Add a note in the figure legend specifying that this figure pertains to cell interactions concerning TAMs.

Response: In the revised figure legend, we have specified that the figure depicts the changes in cell-cell interaction strength between TAMs and other cellular components in the tumors following pericyte-specific sGC inactivation (**Figure EV5A** of the revised manuscript).

Referee #2:

Zhu et al present a study on the contribution of soluble guanylate cyclase (sGC) in pericytes to the tumor microenvironment (TME). The authors demonstrate that loss of sGC in pericytes does not impair vasculature in normal organs but results in modest reduction in tumor growth using three distinct murine tumor cell lines. Extensive scRNAseq analysis of tumors grown in WT or sGC mutant mice indicate that loss of sGC in pericytes alters the TME. The authors details changes in pericyte-EC interaction, TGF β , CAFs and macrophages. The authors then exploit this knowledge pharmacologically by inhibiting sGC with ODQ and demonstrate that ODQ enhances the efficacy of a multi-angioRTKi. Importantly the pharmacologic combination therapy provides more impressive anti-tumor activity than genetic loss of sGC alone. There are multiple interesting aspects of the study; however there are also some challenges as listed below.

Comments:

1. The vascular effects of the loss of sGC in pericytes are interesting.
 - Inclusion of a tracer for functional vasculature would be an important addition to the characterization. Tomato-lectin or similar tools to mark functional vessels is recommended.

Response: We express our gratitude to the reviewer for this important suggestion. To address this point, we perfused the mice with Lectin-FITC (BSI-B4) 15 minutes before tumor resection. Subsequent analysis unveiled a significantly lower vessel perfusion rate in the tumors of $sGC^{\Delta PC}$ mice compared to sGC^{Ctr} mice. We have now incorporated this data as **Figure EV2C** in the revised manuscript.

- The increased albumin and Ter119+ staining in tumors from sGC mutant mice suggests an increase in permeability of the tumor vasculature. Do the areas of elevated albumin correspond to elevated hypoxia? Can the authors provide additional explanation of what they believe is happening?

Response: To address the reviewer's question, we stained hypoxia probe-perfused tumor sections with antibodies against Ter119 and CD31 to examine the correlation between hypoxia and permeability. The presence of Ter119 signal outside of the CD31-positive blood vessels indicates vascular leakage. Our analysis, presented in **Rebuttal Figure 4**, unequivocally demonstrates that elevated vascular permeability is linked to increased hypoxia.

- the Hypoxyprobe data shown in Fig 1H indicates that the edge of the tumor is hypoxic but not the core, this is not typical. An overlay of CD31+ blood vessels with the hypoxyprobe is recommended. Additionally, does 1H shown the entire tumor, if so images showing increased magnification of the edge and central regions of the tumor should be shown.

Response: We apologize for the omission of the CD31 channel in the initial submission.

In response, we have now included the corresponding CD31 staining and incorporated high-magnification images focusing on the vascularization of both the edge and central regions of tumors. The images demonstrate that the areas stained with hypoxia-probe were predominantly located at the periphery of tumors, where vascularization was limited. In contrast, the central regions of the tumors were highly vascularized and showed no hypoxia signal (**Figure 1H**).

2. The concept that loss of sGC in pericytes impacts TGF β activity in the tumor microenvironment is intriguing. However, there are a few things that should be clarified further.

- Figure 4M suggests via bioinformatics means that there is a reduction in TGF β activity in tumors in sGC mutant mice. This should be validated by looking at pSMAD2 or other measures of TGF β activity in tumors by IHC.

Response: Following the reviewer's suggestion, we conducted pSMAD3 staining in the tumors. IHC staining demonstrated a significant reduction in pSMAD3 levels in the tumors of sGC ^{Δ PC} mice compared to sGC^{Ctrl} mice, indicating impaired TGF β activity following pericyte-specific sGC deletion. The data has been discussed in the main text (**line 345-347**) and included in **Figure EV4F** of the revised manuscript.

- Are the levels of TGF β quantified in Fig 4O, total or active TGF β , if total then active should be determined.

Response: Active TGF β was measured in **Fig 4O**, and this has been clarified in the revised figure legend of the methods section.

3. The authors suggest that the data in Fig 4 indicate that CD105 neg apCAFs contribute to the reduced tumor growth of tumors in sGC mutant mice. The data are correlative; there is no functional assessment of CD105 neg CAFs. I strongly recommend softening the language concluding this section. The contribution of apCAFs as tumor suppressive is tentative at best. The current best available data suggests that apCAFs can drive Treg formation in an antigen specific manner (PMID: 35523176) indicating that calling apCAFs tumor suppressive is challenging.

Response: We appreciate the thoughtful feedback provided by the reviewer regarding the interpretation of CD105^{neg} apCAFs' contribution to reduced tumor growth in the current study. In response, we have carefully refined the conclusion in the revised manuscript to convey this aspect more precisely (**line 361-363**).

4. The data in Fig 5 is supportive of the loss of sGC in pericytes impacting myeloid cell phenotype. Please clarify if the CD105^{neg} CAFs used in 5J,K are apCAFs?

Response: We thank the reviewer for this crucial point. Our single-cell analysis unveiled that the predominant fraction of CD105^{neg} CAFs fell within cluster 3, with a small proportion belonging to cluster 2. Cluster 3, markedly expanded in sGC^{ΔPC} tumors, was identified as apCAFs according to their gene expression signature (**Figure EV4A, Figure 4H**). Additionally, qPCR analysis confirmed that CD105^{neg} CAFs expressed higher levels of apCAF-associated genes, such as *Cd74*, *Cd14*, and *Cxcl2*, compared to CD105^{pos} CAFs (**Figure EV4D**). While the data strongly supports that the majority of isolated CD105^{neg} cells were apCAFs, it's important to note that a small fraction corresponds to cluster 2 CAFs (a mixture of iCAFs and myCAFs). Therefore, in Figure 5 J-K, we prefer to refer to these cells as CD105^{neg} CAFs rather than categorizing them directly as apCAFs.

Minor:

1. Supp fig 2 D, merge not shown; Supp fig 9B not all panels shown - I assume these are file transfer/image issues?

Response: We apologize for the errors introduced during combining process. To prevent incomplete figure displaying issues, we have uploaded all individual figures in Tiff format, in addition to the combined PDF file, to ensure proper viewing of the figures.

2. check the wording for the legend of Figure 3, in particular panel H. I believe the color scheme is referenced incorrectly.

Response: We appreciate the reviewer for bringing this mistake to our attention. The error has been corrected, and the figure legend of **Figure 3H** has been rephrased for improved clarity.

3. Supp Fig 9, panel B, the graph y-axis is labeled VEGFR α positive area, this should be PDGFR α

Response: This error has been corrected in the revised manuscript (**Appendix Figure S6**).

4. A suggestion to increase impact. Perform IHC for adhesion molecules (ICAM, VCAM) and T cells in tumors treated with ODQ + Fruquintinib. The data in earlier parts of the manuscript suggest an immune-related mechanism and it is possible that the combo therapy enhances T cell infiltration.

Response: Following the reviewer's suggestion, we performed IHC staining for CD4 and CD8 in tumors treated with Fruquintinib or ODQ + Fruquintinib. Our results showed specific staining for CD4 and CD8, and subsequent analysis revealed an increase in CD4 T cell infiltration upon Fruquintinib treatment compared to the vehicle-treated group. However, the levels of CD4 T cell infiltration upon ODQ + Fruquintinib treatment were similar to those in the Fruquintinib treatment group. Specifically, the levels of infiltrated CD8 T cells remained unaltered between the Fruquintinib-treated group and the group receiving ODQ + Fruquintinib combinatory therapy (**Rebuttal Figure 5A-B**). However, immunofluorescence staining for ICAM1 and VCAM1 showed expression not only in the vascular endothelial cells but also in other cells within the tumors. In line with this, single-cell analysis revealed ICAM1 expression in immune cells and VCAM1 expression in tumor cells (**Rebuttal Figure 5C-D**). Thus, the data could not provide any definitive clues regarding the involvement of T cell infiltration in the potential immune-related mechanisms contributing to the enhanced anti-tumor efficacy of the combination therapy.

References:

Hosaka, K. *et al.* Pericyte-fibroblast transition promotes tumor growth and metastasis. *Proceedings of the National Academy of Sciences* **113**, E5618-27 (2016).

Mayr, D. *et al.* Characterization of the Two Inducible Cre Recombinase-Based Mouse Models NG2-CreERTM and PDGFRb-P2A-CreERT2 for Pericyte Labeling in the Retina. *Curr. Eye Res.* **47**, 590–596 (2022).

Teichert, M. *et al.* Pericyte-expressed Tie2 controls angiogenesis and vessel maturation. *Nature Communications* **8**, 16106 (2017).

Sato, S. *et al.* Mesenchymal Tumors Can Derive from Ng2/Cspg4-Expressing Pericytes with β -Catenin Modulating the Neoplastic Phenotype. *Cell Rep.* **16**, 917–927 (2016).

Vanlandewijck, M. *et al.* A molecular atlas of cell types and zonation in the brain vasculature. *Nature* **554**, 475–480 (2018).

Dear Dr. Hu,

Thank you for submitting a revised version of your manuscript. Your study has now been seen by both original referees, who find that their previous concerns have been addressed and now recommend acceptance of the manuscript.

There now remain a few editorial points that need addressing before I can extend formal acceptance of the manuscript:

1. We are missing the ORCID iD for the co-corresponding author Wu Wei. In order to link the ORCID iD to the account in our manuscript tracking system, the author in question has to do the following:
 - Click the 'Modify Profile' link at the bottom of your homepage in our system.
 - On the next page you will see a box halfway down the page titled ORCID*. Below this box is red text reading 'To Register/Link to ORCID, click here'. Please follow that link: you will be taken to ORCID where you can log in to your account (or create an account if you don't have one)
 - You will then be asked to authorise Wiley to access your ORCID information. Once you have approved the linking, you will be brought back to our manuscript system.Unfortunately, we cannot do this linking on the author's behalf for security reasons.
2. CRediT has replaced the traditional author contributions section because it offers a systematic, machine-readable author contributions format that allows for more effective research assessment. Please remove the Authors Contributions from the manuscript and use the free text boxes beneath each contributing author's name in our online submission system to add specific details on the author's contribution. More information is available in our guide to authors.
3. Please include "Sources of funding" in the Acknowledgments section.
4. Our data editors have flagged the following issues in figure legends that need correcting:
 - Please indicate the statistical test used for data analysis in the legends of figures 2b; 3a; 4c; 5b.
 - Please note that in figures 4o, q; EV 2a-c; there is a mismatch between the annotated p values in the figure legend and the annotated p values in the figure file that should be corrected.
 - Please note that information related to n is missing in the legends of figures 3c, i; 4f, l, n; 5d; EV 5b-c.
 - Please define the scale bar for figures 3d-f, j-k.
5. Please re-organise source data files as one folder per figure. For main figures, please zip files together for each main figure, while for EV and Appendix figures, separate figure folders should be zipped together.
6. Please correct a typo in the synopsis image: should be "Increased CAF population".

Thank you again for giving us the chance to consider your manuscript for The EMBO Journal. I look forward to receiving the final version and your input on the source data issues.

With best wishes,

Ieva

We realize that it is difficult to revise to a specific deadline. In the interest of protecting the conceptual advance provided by the work, we recommend a revision within 3 months (29th May 2024). Please discuss the revision progress ahead of this time with the editor if you require more time to complete the revisions. Use the link below to submit your revision:

Referee #1:

In the revised MS the authors satisfy all this reviewer's comments

Referee #2:

The authors have adequately addressed the prior concerns. I have no other issues with the manuscript.

Authors addressed the remaining issues.

Dear Junhao,

Thank you for addressing the final editorial issues. I am now pleased to inform you that your manuscript has been accepted for publication in the EMBO Journal.

Before we transfer your manuscript to our publishers, I would like to propose minor edits in the synopsis, manuscript title and abstract to increase its accessibility to our more general readership. I have also written a short blurb that will accompany the title of your manuscript in our online table of contents. Please take a look at the text below and in the attached manuscript file and let me know if any adjustments or corrections are needed.

Title:

Pericyte signaling via soluble guanylate cyclase shapes the vascular niche and microenvironment of tumors

Blurb:

Pericyte-specific inhibition of nitric oxide receptor increases the vulnerability of angiogenic tumor vessels to anti-angiogenic therapy.

Synopsis

The precise role and therapeutic targeting potential of pericytes in tumor growth remains insufficiently explored. By selectively disrupting soluble guanylate cyclase (sGC) signaling in pericytes, this study reveals their impact on the vascular niche and tumor microenvironment.

- Pericyte-specific deletion of sGC results in detachment of pericytes from the endothelium within tumors.
- Pericyte sGC deletion induces transcriptome reprogramming in both pericytes and ECs.
- Pericytes and neighboring endothelial cells (ECs) within the vascular niche collectively regulate cancer-associated fibroblasts (CAFs) and tumor-associated macrophages (TAMs) within tumors.
- Inhibition of pericyte sGC enhances the susceptibility of tumor blood vessels to anti-angiogenic therapy.

Finally, we would like to promote your manuscript among the Chinese readership. Therefore, we would like to invite you to prepare a short summary of the manuscript in Chinese (1500-2000 Chinese characters), which we will promote on the WeChat platform 'BioArt' with more than 610,000 followers.

If you are interested in this opportunity, we recommend covering the article very close to its online publication date. Thus, ideally we would very much appreciate if you could send us a draft within the next 7 working days. Please let us know whether or not you would be interested in contributing such a short summary in Chinese.

I have included below some general guidelines on how to prepare a summary and a link to recent examples for your reference. Please let me know if you have any questions about this.

If you have any questions, please do not hesitate to contact the Editorial Office. Thank you for this contribution to The EMBO Journal and congratulations on a nice study!

Best wishes,

Ieva

Ieva Gailite, PhD
Senior Scientific Editor
The EMBO Journal

Meyerhofstrasse 1
D-69117 Heidelberg
Tel: +4962218891309
i.gailite@embojournal.org

General WeChat Summary Guidelines

1. These summary articles are meant to be targeting general audience, so please limit the use of specialized technical terms, acronyms and jargon.
2. A summary usually starts with brief background information of the reported work, which is followed by explaining the findings in some detail, and ends with a short review of the conclusions as well as the implications of the work and future directions for the research.
3. The summary should at least contain one graphical item, such as a scheme or a figure from the paper.
4. Please provide ONE SINGLE document containing all text and graphical materials, ideally as a Word.docx or .doc file. Please DO NOT provide the document as a .pdf file.
5. Please DO NOT publicly release the document before the paper is officially published online.

Summary Examples

EMBO J | 罗招庆/欧阳松应揭示谷酰胺脱氨酶MvcA的去泛素化功能

EMBO J | 王松灵院士团队揭示组织内应力调控大型哺乳动物乳恒牙替换的新机制
